# Preventing Differential Cryptanalysis Attacks Using a KDM Function and the 32-Bit Output S-Boxes on AES Algorithm Found on the Internet of Things Devices

**Khumbelo Difference Muthavhine * and Mbuyu Sumbwanyambe ***

Department of Electrical and Mining Engineering, University of South Africa, Johannesburg 2000, South Africa
* Correspondence: kdmuthavhine@gmail.com (K.D.M.); sumbwm@unisa.ac.za (M.S.)

**Abstract:** Many Internet of Things (IoT) devices use an Advanced Encryption Standard (AES) algorithm to secure data stored and transmitted during the communication process. The AES algorithm often suffers DC (DC) attacks. Little has been done to prevent DC attacks, particularly on an AES algorithm. This study focuses on preventing Differential Cryptanalysis attacks. DC attacks are practiced on an AES algorithm that is found on IoT devices. The novel approach of using a Khumbelo Difference Muthavine (KDM) function and changing the $8 \times 8$ S-Boxes to be the $8 \times 32$ S-Boxes successfully prevents DC attacks on an AES algorithm. A KDM function is a newly mathematically developed function, coined and used purposely in this study. A KDM function was never produced, defined, or utilized before by any researcher except for in this study. A KDM function makes a new 32-Bit S-Box suitable for the new Modified AES algorithm and confuses the attacker since it comprises many mathematical modulo operators. Additionally, these mathematical modulo operators are irreversible. The study managed to prevent the DC attack of a minimum of 70% on AES and a maximum of 100% on a Simplified DES. The attack on the new Modified AES Algorithm is 0% since no S-Box is used as a building block.

**Keywords:** Internet of Things (IoT); DC attacks; Advanced Encryption Standard (AES); $8 \times 8$ S-Boxes; $8 \times 32$ S-Boxes; Khumbelo Difference Muthavine (KDM) function; Internet of Things (IoT) devices





## 1. Introduction

IoT devices and platforms are advancing boundless while initiating a seamless combination of computer networks with things or objects [1,2]. IoT is an open network platform and a new communication standard for the latest innovations, connecting multiple heterogeneous devices to render new conventional services [3,4]. Nonetheless, the tremendous benefits of utilizing IoT devices face diverse predicaments to solve and reach IoT full adoption. Security and privacy are the crucial predicaments for the IoT devices and yet admit some of the immense inconveniences such as DC attacks [3,4]. IoT devices and platforms, with no skepticism, depend on cryptographic algorithms such as AES for the security and privacy of confidential information and data [3,4].

Consequently, new services provided by IoT devices have to be sufficiently secured utilizing solid cryptographic algorithms such as AES [1,2]. A cryptographic algorithm is a mathematical method that converts plaintext (simple messages) into ciphertext (unclear messages), and vice versa [5,6]. At the same time, while an improvement of security and privacy on IoT devices is observed, there is also an increasing use of old cryptographic algorithms such as AES. The attackers create and improve different techniques of attacking the distinct solid algorithms [7,8]. The most standard algorithms, such as AES, are being attacked using various mathematical methods, such as DC attacks [1,9]. For instance, four-round AES can be attacked using the DC attacks [1]. DC attacks are subjected to a differential that is supplemented by a significant probability [1]. AES has been implemented on other IoT devices to secure data used for online transactions such as smart cards [1].

AES has been attacked using the DC attacks on a reduced number of rounds version, and the complexity has been compared to that of an exhaustive research attack [10]. DC attacks have worked successfully and faster than comprehensive research attacks, which have been treated as the upper bound attack in cryptography [6,10].

An AES algorithm is still used to secure confidential information and data stored in IoT devices today [3,4]. For instance, cell phones and wireless networks as IoT devices are using AES for confidentiality, integrity, and availability of data [3]. Additionally, the packet filtering methods of cellular phones and wireless networks are using AES for security [3]. In addition, AES is used on IoT sensors where communication security is being established in various IoT devices such as intelligent energy-grids, Machine to Machine (M2M) communications, buildings, and data computing devices [11]. IoT boards, the CYW board, IoT edge, and BCM board, as examples of IoT devices, are using an AES algorithm for data security [12]. The PRISEC module of the UbiPri middleware is one of the IoT devices that have been using an AES to secure data privacy and protection [13].

This study focuses essentially on the DC attacks of an AES commonly encountered and required in IoT devices. DC attacks of an AES are the biggest problem on confidential information and data. An intruder can effortlessly attack an AES because of fewer output bits from an AES S-Box and its inverse. An AES needed in IoT devices has 8-output bits from the S-Boxes, far less than 32 bits.

A newly generated 32 output bits S-Box prevents DC attacks of an AES found on IoT devices. A KDM function makes a new 32-Bit S-Box suitable for the new Modified AES algorithm, which confuses the attacker. The novel approach of using a KDM function and changing the $8 \times 8$ S-Boxes to be the $8 \times 32$ S-Boxes successfully blocks DC attacks on an AES algorithm. A KDM function is a new mathematically developed function, coined and used purposely in this study. A KDM function was never produced, defined, or utilized before by any researcher except for in this study.

The principal concern of this study is a DC attack practiced on IoT devices by trespassers to identify the cryptographic keys of an AES algorithm. An AES can suffer a DC attack [1,9]. For instance, a DC attack was implemented experimentally on a Mini-AES algorithm [14]. The experiment exposed more than 50 percent of the secret key. In addition, an AES was attacked, utilizing an algebraic DC attack to decode the secret key [15]. The basic principle of the DC attack adventured the high probability of appropriate events of plaintext pair differences and ciphertext pair differences created in the decisive round [7]. Lacko-Bartosova [16] showed a DC attack of a two-round AES with a complexity approach of a three-round AES attack. Lacko-Bartosova [16] also showed that a DC attack depends on the support of extraordinary bitwise text differences. Grassi [5] attacked a five-round AES utilizing a DC attack and "multiple-of-8" rule. Tunstall [1] says the first attack is a four-round AES DC attack controlled to a differential that completed a significant probability. The second attack was a five-round AES Square attack that required a time complexity of 237.5 throughout the encryption process and 28 pairs of ciphertexts to crack an AES secret key [1].

IoT devices use an AES algorithm to encrypt and convey the encrypted data to the next layer of security, which is known as the Message Queuing Telemetry Transport protocol [4]. The Message Queuing Telemetry Transport protocol is an ISO standard (ISO/IEC PRF 20922). They are then used to transfer encrypted data. On the receiver side, the encrypted data was decrypted using an AES algorithm [4]. The VMware SD-WAN Edge holds. The VMware SD-WAN Dynamic Multipath Optimization (DMPO) and an Extensive Application Recognition as IoT devices aggregated on reoccurring links related to regulating traffic across optimal links [17]. Additionally, traffic is being directed to other VMware SD-WAN Edges of distinct departments, private data centers, universities, and offices, utilizing an AES for secure communication [17]. Sophia et al. [18] showed that the health department is a growing concern for patients worldwide. An e-healthcare Remote Clinical Sensor Network is supported in accumulating the vital body information

of personal terminals using sensors as IoT devices. The recommended technique was for policies executing a secured key and encoded by an AES [18].

With all this knowledge, the interest of this study is to recure an AES from DC attacks and secure all IoT devices utilizing an AES algorithm. A DC attack can destroy the complete security of IoT devices and users if it is not appropriately examined. Little has been conducted to advance the number of output bits on the S-Boxes to combat a DC attack [1,9]. This study concentrates on retaliating a DC attack on an AES.

The newly generated 32 output bits S-Boxes are employed to obstruct the DC attacks of an AES identified on IoT devices. A KDM function makes a new 32-Bit S-Box suitable for the new Modified AES Algorithm and confuses the attacker since it comprises many mathematical modulo operators. Additionally, most mathematical modulo operators are irreversible. Additionally, the novel approach of applying a KDM function and transforming the 8 × 8 S-Boxes to be the 8 × 32 S-Boxes successfully blocks DC attacks on an AES algorithm. A KDM function is a new mathematically function, generated, named, and used purposely for this study. A KDM function was never developed, defined, explained, or employed before by any researcher besides in this study.

*1.1. An AES Algorithm*

AES algorithm is a symmetrical cryptographic algorithm, which is widely and commonly applied in IoT devices with a block size of 128-Bit [19,20]. An AES has four main steps, called functions, namely: *Substitute Byte* (*SubByte*), *Shift Rows* (*ShiftRows*), *Mix Columns* (*MixColumn*), and finally *Add Round Key* (*AddRoundKey*) [19,20]. With these four main functions, three functions have inverses, namely: *Inverse Mix Columns* (*InvMixColumn*), *Inverse Substitute Byte* (*InvSubByte*), and *Inverse Shift Rows* (*InShiftRows*). The *Add Round Key* (*AddRoundKey*) is the only function that does not have an inverse [2,19]. The main functions are employed during the encryption process, and inverses are employed during the decryption process [19,20]. Figure 1 depicts the encryption and decryption processes. During the encryption process, the initial step or function is *SubByte*. In this function, an AES algorithm uses a Substitution-Box (S-Box). An S-Box is a look-up table comprised of inputs and outputs in the number of bytes [2,19]. In the *SubBytes* step, each input byte is replaced by a different unconventional byte using an AES S-Box [19,20]. Referring to Figure 2, assume that the input byte is *c*000 in hexadecimal notation, *c*0 = *x*, which is a row number, and 00 = *y*, which is a column number. Examining from an AES S-Box on Figure 2 where *x* and *y* intersect, *c*000 is replaced by *ba*. During the decryption process, an inverse AES S-Box is employed. When an inverse AES S-Box is employed, the step is called *InvSubBytes*, step number three during the decryption process. The *InvSubByte* is a straight inverse of the *SubByte*. Referring to Figure 2. An AES changes a string of plaintext (input) into 4×4 matrix; after the replacement or substitution, the matrix is called the state of an AES. Note that a state is referred to the output of each step or function of an AES. Another critical function that operates the state is *MixColumns*. The mixing or *MixColumns* is the multiplication method of mixing matrix rows and columns. Each 8-Bit entity of a row is multiplied by each 8-Bit entity of the state column using matrix transformation. In simple terms, each row of the matrix transformation is employed to multiply every column of the state [19,20]. The outputs of multiplication are XORed to produce a distinct state. The reverse transformation of *MixColumn* is called *InvMixColums*. *InvMixColums* is achieved during the decryption process [2,19]. The size of states is constantly the same size, which is a 4×4 matrix. Refer to Figure 3.

The last function or step of an AES during the encryption process is called *Add Round Key* (*AddRoundKey*). Unlike other functions, the *AddRoundKey* does not have an inverse. The method of the *AddRoundKey* is implemented to both the encryption and decryption process. During the *AddRoundKey* operation, either the state produced after *MixColumns* or *InvMixColums* are XORed with the state of key [2,19]. For detail, refer to Figure 4.

An AES supports three original sizes of keys, namely: 192-Bit, 128-Bit, and 256-Bit [19,20]. The encryption process involves 10 rounds of altering for 128-Bit key, 14 rounds for 256-Bit key, and 12 rounds for 192-Bit key [2,19]. All subkeys are produced from an initial key; producing subkeys depends on the size of the initial key. Subkeys are used during encryption and decryption processes [19,20]. The mathematical steps explaining the generation of subkeys are given in Figure 5.

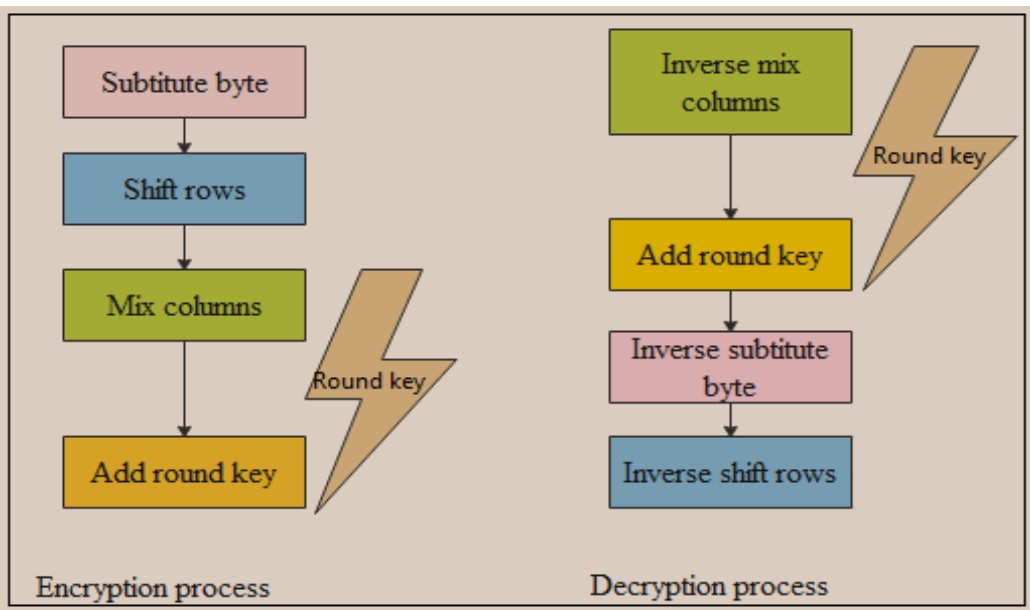

**Figure 1.** Encryption and decryption processes of an AES.

### 1.2. DC Attack

A DC attack utilizes the high probability of specific events of plaintext differences and differences into the final round of the algorithm [1,9]. For instance, consider an algorithm with input (plaintext) $P = [P_1, P_2, ..., P_n]$ and output (ciphertext) $C = [C_1, C_2, ..., C_n]$ [7]. Suppose that two inputs to the algorithm are $P'$ and $P''$ with the complementary outputs $C'$ and $C''$, respectively. The input difference is calculated by $\Delta P = P' \oplus P''$, the symbol $\oplus$ indicates XOR bitwise operator, and hence $\Delta P_i = P'_i \oplus P''_i$, correspondingly to the output difference where, $\Delta C = C' \oplus C''$ and $\Delta C_i = C'_i \oplus C''_i$ [8]. The intruder has to find the high differential probabilities of each S-Box utilized in the particular algorithm to implement a DC attack [1,9]. Then the intruder calculates outputs of high differential probabilities of S-boxes, which affect the known-plaintext difference $\Delta P = P' \oplus P''$ corresponding to the ciphertext difference $\Delta C = C' \oplus C''$ [7,8]. Additionally, the intruder constructs the Difference-Distribution tables for each S-Box for input difference $\Delta P$ and output difference $\Delta C$ to discover the differential characteristic. Many S-Boxes used by the different algorithms are weak due to the size of both input and output bits [1,7]. Regarding an S-Box's weakness, the intruder may easily calculate the high difference probabilities of pair $(\Delta P_i, \Delta C_i)$ of $(1/(2^n))$, where $n$ is the number of bits used as an output [7,8]. The intruder analyses all different pairs of input $P_i$ and output $C_i$ of an S-Box, where $i$ represents the $i-th$ bit of the $P_i$ and $C_i$, respectively. The high difference probabilities of pair $(\Delta P_i, \Delta C_i)$ of each S-Boxes are combined and used from the first round to the second last round, utilizing the S-Boxes as an independent building block of the particular algorithm. Suppose that the differential characteristic for the second last round gives a desirable high enough probability $pD$. In that case, it is easy to discover certain bits of the key or subkey used on the last round subkey by XORing all the potential keys of all affected non-zero difference bits TPS (Target Partial Subkeys) utilizing the last round with the output and operating one round backward through the S-Boxes. The number of known plaintext–ciphertext pair differences needed for the intruder is $1/pD$ [7,8].

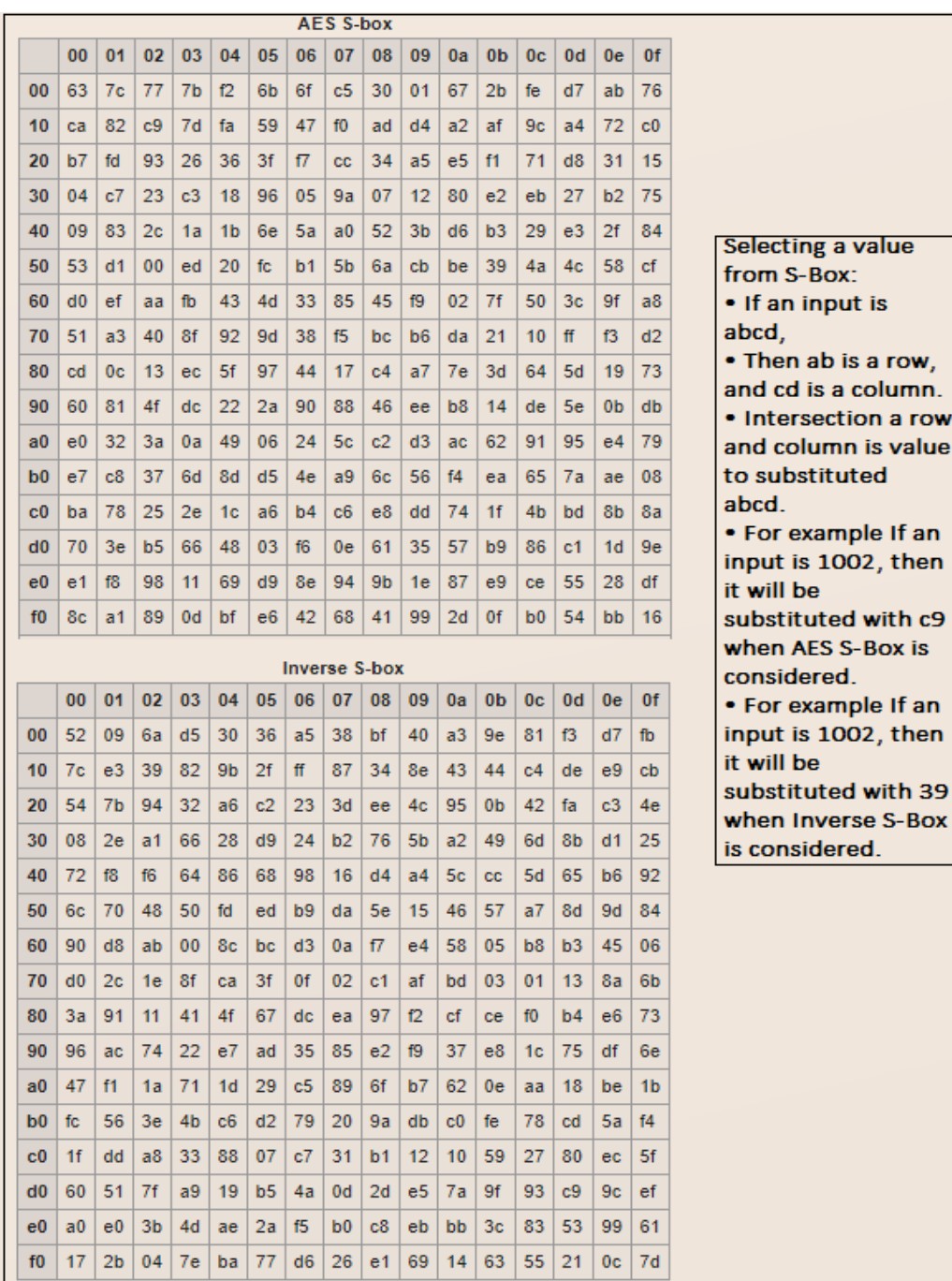

**Figure 2.** SubByte and InveSubBytes of an AES with example.

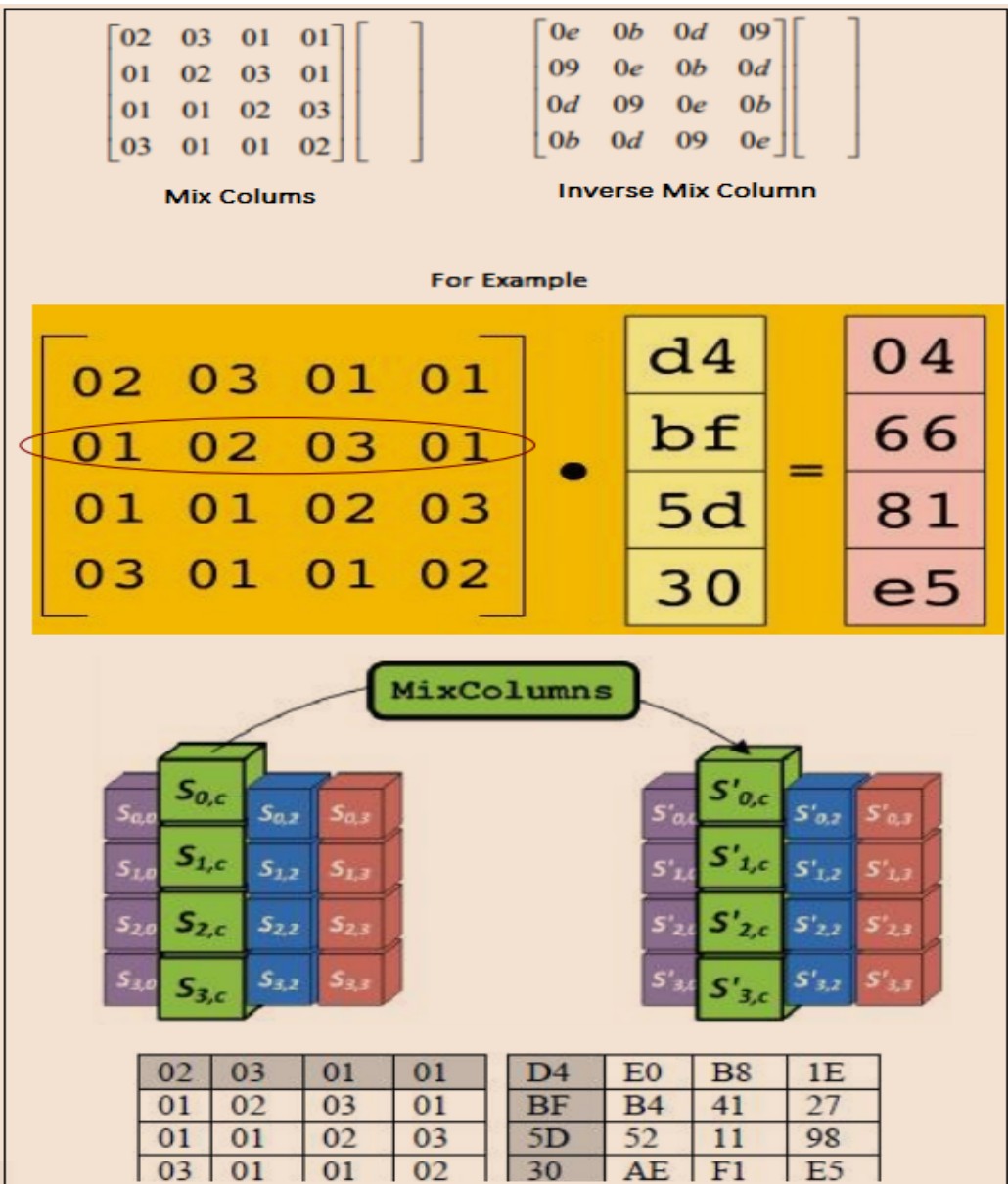

**Figure 3.** Mix columns and inverse mix columns of an AES.

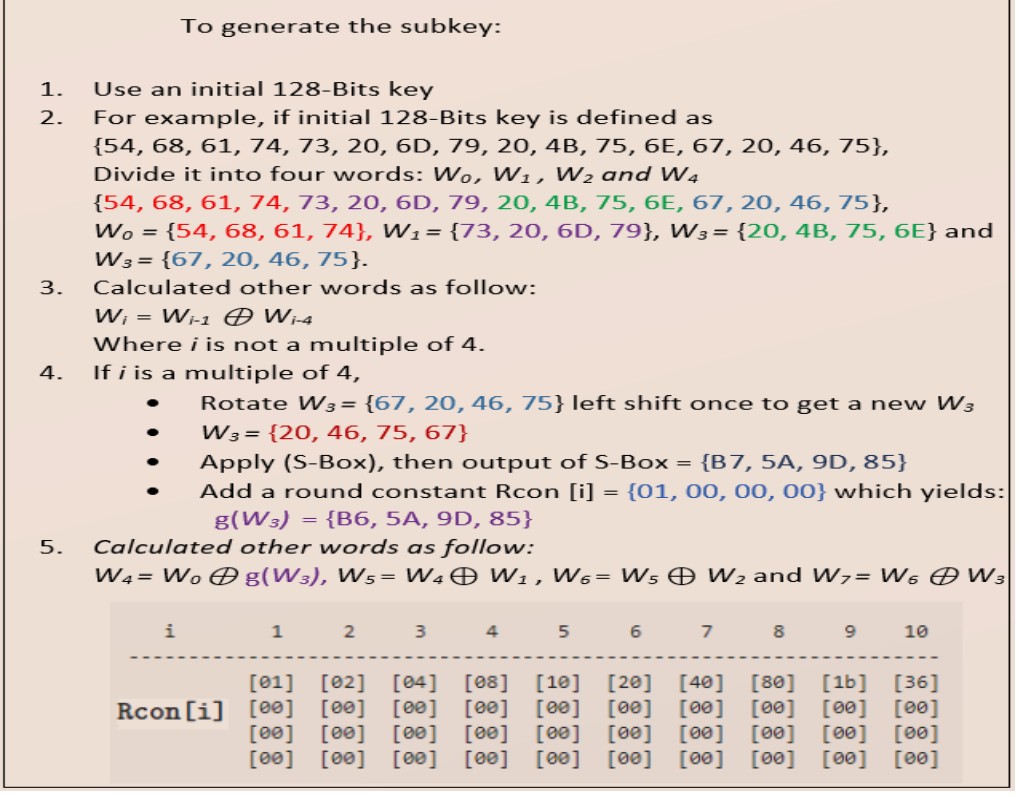

**Figure 4.** Adding key process of an AES.

To generate the subkey:

1. Use an initial 128-Bits key
2. For example, if initial 128-Bits key is defined as
   {54, 68, 61, 74, 73, 20, 6D, 79, 20, 4B, 75, 6E, 67, 20, 46, 75},
   Divide it into four words: $W_0$, $W_1$, $W_2$ and $W_4$
   {54, 68, 61, 74, 73, 20, 6D, 79, 20, 4B, 75, 6E, 67, 20, 46, 75},
   $W_0$ = {54, 68, 61, 74}, $W_1$ = {73, 20, 6D, 79}, $W_3$ = {20, 4B, 75, 6E} and
   $W_3$ = {67, 20, 46, 75}.
3. Calculated other words as follow:
   $W_i = W_{i-1} \oplus W_{i-4}$
   Where *i* is not a multiple of 4.
4. If *i* is a multiple of 4,
   - Rotate $W_3$ = {67, 20, 46, 75} left shift once to get a new $W_3$
   - $W_3$ = {20, 46, 75, 67}
   - Apply (S-Box), then output of S-Box = {B7, 5A, 9D, 85}
   - Add a round constant Rcon [i] = {01, 00, 00, 00} which yields:
     $g(W_3)$ = {B6, 5A, 9D, 85}
5. *Calculated other words as follow:*
   $W_4 = W_0 \oplus g(W_3)$, $W_5 = W_4 \oplus W_1$, $W_6 = W_5 \oplus W_2$ and $W_7 = W_6 \oplus W_3$

| i | 1 | 2 | 3 | 4 | 5 | 6 | 7 | 8 | 9 | 10 |
|---|---|---|---|---|---|---|---|---|---|----|
| | [01] | [02] | [04] | [08] | [10] | [20] | [40] | [80] | [1b] | [36] |
| Rcon[i] | [00] | [00] | [00] | [00] | [00] | [00] | [00] | [00] | [00] | [00] |
| | [00] | [00] | [00] | [00] | [00] | [00] | [00] | [00] | [00] | [00] |
| | [00] | [00] | [00] | [00] | [00] | [00] | [00] | [00] | [00] | [00] |

**Figure 5.** Key scheduling of an AES.

In a DC attack, the intruder examines the difference pairs of the S-Boxes found in the cryptographic algorithm. For instance, suppose a $4 \times 4$ S-Box was illustrated in Table 1 with plaintext $P = [P_1, P_2, P_3 P_4]$ and ciphertext $C = [C_1, C_2, C_3 C_4]$ [7,8]. All difference pairs of an S-Box illustrated in Table 1, $(\Delta P_i, \Delta C_i)$, can be scrutinized and the

probability of $\Delta C_i$ given $\Delta P_i$ can be calculated by considering ciphertext pairs $(P', P'')$ such that $\Delta P = P' \oplus P''$ [1,21]. For a $4 \times 4$ S-Box like the one illustrated in Table 1 the intruder only considers all $16 = (2^4)$ values for $P'$ and then the value of $\Delta P_i$ shows the value of $P''$ to be $P'' = P' \oplus \Delta P$ [7,8].

Considering a $4 \times 4$ S-Box illustrated in Table 1, the intruder can calculate the probability values of $\Delta C$ for each plaintext pair $(P', P'' = P' \oplus \Delta P)$ [1,22]. For instance, the binary values of $P$, $C$, and the ciphertext values for $\Delta C$ for given plaintext pairs $(P, P \oplus \Delta P)$ are presented in Table 2 for $\Delta P$ values of $1011_{binary\ number}$, $1000_{binary\ number}$, and $0100_{binary\ number}$. The last three columns of Table 2 depict $\Delta C$ values for the P value row and the particular $\Delta P$ value column [1,15]. From Table 2, the intruder can observe that the occurrence number of $\Delta C = 0010_{binary\ number}$ for $\Delta P = 1011_{binary\ number}$ is 8 over 16 possible values, then the probability = 8/16; the occurrence number of $\Delta C = 1011_{binary\ number}$ given $\Delta P = 1000_{binary\ number}$ is 4 over 16; the occurrence number of $\Delta C = 1_{binary\ number}$ given $\Delta C = 0100_{binary\ number}$ is 0 over 16 [1,9].

The intruder tabularizes the entire data for a $4 \times 4$ S-Box illustrated in Table 1 in a Difference–Distribution Table in which the columns represent $\Delta C_{hexadecimal}$ and the rows represent $\Delta P$ values [8,22]. The Difference-Distribution Table for a $4 \times 4$ S-Box illustrated in Table 1 is given in Table 3 [5,7]. Each element of Table 3 depicts the occurrence number of the corresponding ciphertext difference $\Delta C$ value given the plaintext difference $\Delta P$ [1,8,10,16]. The intruder can observe that, besides the specific cases of $(\Delta P = 0, \Delta C = 0)$, the highest value in Table 1 is 8, corresponding to $\Delta P = B_{hexidecimal}$ and $\Delta C = 2_{hexidecimal}$ [1,21]. In consequence, the probability that $\Delta C = 2_{hexidecimal}$ knowing an arbitrary pair of plaintext values that satisfy $\Delta P = B_{hexidecimal}$ is 8/16 [7,8]. On the contrary, the smallest value in Table 1 is 0 and happens for various difference pairs. In this situation, the probability of the $\Delta C$ value happening knowing the $\Delta P$ value is 0. With all this information on hand, the intruder can simply discover the highest percentage of secret bits key of any algorithm using a similar S-Box like the one defined in Table 1 [7,8]. The few remaining bits of the secret key are found using simple mathematical and statistical analysis and a trial and error method.

**Table 1.** A simplified DES's S-Box.

| P | 0 | 1 | 2 | 3 | 4 | 5 | 6 | 7 | 8 | 9 | A | B | C | D | E | F |
|---|---|---|---|---|---|---|---|---|---|---|---|---|---|---|---|---|
| S(P) = C | 4 | E | D | 1 | 2 | F | B | 8 | 3 | A | 6 | C | 5 | 9 | 0 | 7 |

**Table 2.** Representation of difference pairs of a 4 × 4 S-Box.

| P | C | $\Delta C$<br>$\Delta P = 1011$ | $\Delta C$<br>$\Delta P = 1000$ | $\Delta C$<br>$\Delta P = 0100$ |
|---|---|---|---|---|
| 0000 | 1110 | 0010 | 1101 | 1100 |
| 0001 | 0100 | 0010 | 1110 | 1011 |
| 0010 | 1101 | 0111 | 0101 | 0110 |
| 0011 | 0001 | 0010 | 1011 | 1001 |
| 0100 | 0010 | 0101 | 0111 | 1100 |
| 0101 | 1111 | 1111 | 0110 | 1011 |
| 0110 | 1011 | 0010 | 1011 | 0110 |
| 0111 | 1000 | 1101 | 1111 | 1001 |
| 1000 | 0011 | 0010 | 1101 | 0110 |
| 1001 | 1010 | 0111 | 1110 | 0011 |
| 1010 | 0110 | 0010 | 0101 | 0110 |
| 1011 | 1100 | 0010 | 1011 | 1011 |
| 1100 | 0101 | 1101 | 0111 | 0110 |
| 1101 | 1001 | 0010 | 0110 | 0011 |
| 1110 | 0000 | 1111 | 1011 | 0110 |
| 1111 | 0111 | 0101 | 1111 | 1011 |

**Table 3.** Difference-distribution table.

| Input Difference<br>$\Delta P$ | Output Difference $\Delta C$ | | | | | | | | | | | | | | | |
|---|---|---|---|---|---|---|---|---|---|---|---|---|---|---|---|---|
| | **0** | **1** | **2** | **3** | **4** | **5** | **6** | **7** | **8** | **9** | **A** | **B** | **C** | **D** | **E** | **F** |
| 0 | 16 | 0 | 0 | 0 | 0 | 0 | 0 | 0 | 0 | 0 | 0 | 0 | 0 | 0 | 0 | 0 |
| 1 | 0 | 0 | 0 | 2 | 0 | 0 | 0 | 2 | 0 | 2 | 4 | 0 | 4 | 2 | 0 | 0 |
| 2 | 0 | 0 | 0 | 2 | 0 | 6 | 2 | 2 | 0 | 2 | 0 | 0 | 0 | 0 | 2 | 0 |
| 3 | 0 | 0 | 2 | 0 | 2 | 0 | 0 | 0 | 0 | 4 | 2 | 0 | 2 | 0 | 0 | 4 |
| 4 | 0 | 0 | 0 | 2 | 0 | 0 | 6 | 0 | 0 | 2 | 0 | 4 | 2 | 0 | 0 | 0 |
| 5 | 0 | 4 | 0 | 0 | 0 | 2 | 2 | 0 | 0 | 0 | 4 | 0 | 2 | 0 | 0 | 2 |
| 6 | 0 | 0 | 0 | 4 | 0 | 4 | 0 | 0 | 0 | 0 | 0 | 0 | 2 | 2 | 2 | 2 |
| 7 | 0 | 0 | 2 | 2 | 2 | 0 | 2 | 0 | 0 | 2 | 2 | 0 | 0 | 0 | 0 | 4 |
| 8 | 0 | 0 | 0 | 0 | 0 | 0 | 2 | 2 | 0 | 0 | 0 | 4 | 0 | 4 | 2 | 2 |
| 9 | 0 | 2 | 0 | 0 | 2 | 0 | 0 | 4 | 2 | 0 | 2 | 2 | 2 | 0 | 0 | 0 |
| A | 0 | 2 | 2 | 0 | 0 | 0 | 0 | 0 | 6 | 0 | 0 | 2 | 0 | 0 | 4 | 0 |
| B | 0 | 0 | 8 | 0 | 0 | 2 | 0 | 2 | 0 | 0 | 0 | 0 | 0 | 2 | 0 | 2 |
| C | 0 | 2 | 0 | 0 | 2 | 2 | 2 | 0 | 0 | 0 | 0 | 2 | 0 | 6 | 0 | 0 |
| D | 0 | 4 | 0 | 8 | 0 | 0 | 0 | 4 | 2 | 0 | 2 | 0 | 2 | 0 | 2 | 0 |
| E | 0 | 0 | 2 | 4 | 2 | 0 | 0 | 0 | 6 | 0 | 0 | 0 | 0 | 0 | 2 | 0 |
| F | 0 | 2 | 0 | 0 | 6 | 0 | 0 | 0 | 0 | 4 | 0 | 2 | 0 | 0 | 2 | 0 |

*1.3. A KDM Function*

In this study, a new function called a KDM function is included. Refer to Figure 6. A KDM function is a newly generated C++ function applied only to intensify a DC attack blockage on an AES algorithm needed on IoT devices. This function is acquired after the S-Boxes of an AES algorithm are modified to generate the 32-bits output S-Boxes. The main function of a KDM function is to assure that the newly 32-bits output S-Boxes fit an AES

algorithm infrastructure. In simple terms, a KDM function coordinates all newly 32-bits output S-Boxes to be efficiently used throughout the encryption and decryption process of the newly adjusted AES algorithm. A KDM function is used to make a new 32-Bit S-Box suitable for the new modified AES algorithm and confuse the attacker since it comprises many mathematical modulo operators. Additionally, most mathematical modulo operators are irreversible. Without a KDM function, the newly generated 32-bits output S-Boxes will not be set in algorithms. This KDM function has particular properties to assure that a DC attack is blocked. These properties are:

1.  The output of a KDM function is not determined, unlike in the S-Boxes, where a look-up table is applied with determined inputs and outputs;
2.  The output of a KDM function is secret and calculated, unlike in an AES S-Boxes, where the output is noticeable on a look-up table;
3.  A KDM function is unchangeable. If one identifies an output of a KDM function, that does not mean an input can be reversely calculated and recovered. The reason is that a KDM function is comprised of several quantities of modular operators;
4.  Chosen constant numbers (such as *Muthavine*, *Khumbelo*, and *Difference*) used in a KDM function are un-factorizable. Refer to Figure 6;
5.  All functions utilized to comprise a KDM function are non-linear;
6.  The input of a KDM function is 32-bits long, and the attacker can not simply construct the Difference-Distribution Table of $2^{32}$ using a computer;
7.  A KDM function accepts the output of the 32-bits S-Boxes and handles it as its input. Then, a distinct output value is created to be applied in the modified AES algorithm. A new particular output value is unpredictable; hence it brings confusion to the attacker;
8.  The output of the 32-bits S-Boxes is defined as *state32hold*. The KDM function takes this output as its input and returns an unpredicted variable called *Khumbelo*. Refer to Figure 6;
9.  After implementing a KDM function, all functions in an AES algorithm calling the S-Boxes have to call or use a KDM function because S-Boxes are mathematically protected and unchangeable in a KDM function;
10. A KDM function makes the 32-Bit output S-Boxes tamper-proof. If the positions of the 32-Bit output S-Boxes are changed, or the 32-Bit S-Boxes are replaced, then M_AES will not yield the expected results.

This study uses a KDM function to make a new 32-Bit S-Box suitable for the new Modified AES Algorithm and confuse the attacker since it comprises many mathematical modulo operators. Additionally, most mathematical modulo operators are irreversible. A KDM function has added robustness against a DC attack, unlike the conventional S-Boxes utilized in predominant AES algorithms. A KDM function operates successfully in both the appropriateness of the newly 32 bits S-Boxes and in the blocking of a DC attack of a recently modified AES algorithm. Mathematically, a KDM function is constructed as follows:

Assign: *Muthavhine* = 4294967296, *Khumbelo* = 4559351687 and *Difference* = 4302746963

Create the first *for* both *i* and *j* less than four, where both *i* and *j* range form 0 to 4, do: assign

$T = state32hold[j][i] \times \left(\frac{state32hold[j][i]}{Muthavhine}\right)$. where $state32hold[j][i]$ is an input of a KDM function from a 32-bit S-Box.

do: assign

$V = Muthavhine \times \left(\frac{Muthavhine}{state32hold[j][i]}\right)$

Change the value of $state32hold[j][i]$, to be the value of $T + V$ by assigning $state32hold[j][i] = T + V$.

Close the first *for* loop.

Create an array of six elements called $Arraof6$ and assign to as $Arraof6 = 256604724$, 40037230360, 7779667, 4294968531, 0273, 4 where $Arraof6_0$ is the first element of $Arraof6$ defined as $Arraof6_0 = 256604724$, $Arraof6_1 = 40037230360$, ..., $Arraof6_5 = 4$.

Create the second *for* loop both $i$ and $j$ less than four, where both $i$ and $j$ range form 0 to 4.

Recall the value of $state32hold[j][i]$ calculated from the *first* for loop.

Compare the value of $state32hold[j][i]$ to the value of $Muthavhine$.

Create condition one: if $state32hold[j][i]$ is greater than $Muthavhine$, then do: assign

$Khumbelo = Arraof6_0 \oplus Khumbelo$

$Difference = (Arraof6_2 \oplus Muthavhine) \, modulo \, (Khumbelo)$.

Where *modulo* operation is the mathematical operator that returns the remainder of a division $(Arraof6_2 \oplus Muthavhine)$ divided by $Khumbelo$.

do: assign

$Muthavhine = (Arraof6_2 \oplus Difference) \, modulo \, (Arraof6_3)$.

Close condition one.

Recall the value of $state32hold[j][i]$ calculated from the *first* for loop.

Compare the value of $state32hold[j][i]$ to the value of $Muthavhine$.

Create condition two: if $state32hold[j][i]$ is less than or equal to $Muthavhine$, then do: assign

$Muthavhine = (state32hold[j][i] <<< Arraof6_4) \, modulo \, (Khumbelo)$.

Where $<<<$ is left circular shifting of the bits, for instance, 5 in decimal = 0101 in binary. If 0101 is left-shifted by 1, then 0101 will be 1010 in binary, which equals 10 in decimal or $A$ in hexadecimal.

do: assign

$Khumbelo = (state32hold[j][i] <<< Arraof6_5) \, modulo \, (Difference)$.

$Difference = (state32hold[j][i] \, modulo \, Khumbelo)$

$<<< Arraof6_4)$.

$Khumbelo = (Muthavhine \oplus Difference) \, modulo \, (Arraof6_2)$.

$Difference = Muthavhine \oplus Khumbelo + Arraof6_0)$.

$Muthavhine = (Khumbelo \oplus Difference) \, modulo \, (Muthavhine)$.

Close condition two and the second *for* loop.

Create the third *for* loop where $i$ and $j$ are less than four, where both $i$ and $j$ range for 0 to 4.

Recall all the returned values calculated from the first and second *for* loops. If the value returns to variable $Khumbelo$, greater than 0, then create a variable $TempState$.

do: assign

$TempState = NOT(state32hold[j][i]) AND Khumbelo$.

Where $NOT$ and $AND$ are bitwise operators. Note that $NOT$ return negative number increased by 1 if an input is a positive integer. For instance, $NOT(2) = -3$, $NOT(5) = -6$, $NOT(10) = -11$ and so on.

do: assign

$state32hold[j][i] = |(state32hold[j][i] \oplus Khumbelo|$, where $|x|$ means absolute operator. An absolute operator changes every negative value to be positive. For instance, $|-x| = |x| = x$.

do: assign

$statehold[j][i] = (\frac{state32hold[j][i]}{Arraof6_2 \oplus Muthavhine}) \oplus Mod4$.

do: assign

$Khumbelo = TempState <<< 1$.

Note that the expression of $Khumbelo = TempState <<< 1$ always reduces the value of $Khumbelo$ until $Khumbelo$ is less than 0. It also checks if $Khumbelo$ is greater than 0. If $Khumbelo$ is greater than 0, repeat the third *for* loop repeated until $Khumbelo$ is less than 0.

Else do: assign

$TempState = Khumbelo \oplus TempState$

$Khumbelo = Khumbelo(\, modulo \, (Muthavhine))$

Send or return the new value of $statehold[j][i]$ to be used by other AES functions or building blocks

Close the third *for* loop.

Close a KDM function.

A KDM Function takes 32-bit output value from an S-Box as $state32hold[j][i]$ and returns a new value $state32hold[j][i]$ value as an output. A KDM Function also makes *Muthavhine* value, *Difference* value, and *Khumbelo* value be un-factorizable polynomials, then modular operators are used for confusion and diffusion to block reverse engineering for intruders. The modular operator (*modulo*) changes the value of the variables inside a KDM Function. The modular operator also gives a confusion range of input when intruders reverse back a KDM Function to guess the correct information used in that event. The value of *Muthavhine*, *Difference*, and *Khumbelo* also constantly kept un-factorizable polynomial variables non-linear and cumbersome in order to construct a Difference Distribution Table using any machine. Modular operators also make variables unknown, invisible, and irreversible to intruders. A KDM function makes a new 32-Bit S-Box suitable for the new Modified AES Algorithm and confuses the attacker since it comprises many mathematical modulo operators. Additionally, most mathematical modulo operators are irreversible. For more mathematical features of a KDM function and C++ comments, refer to Figure 6. For more detail of a KDM function and flowchart, refer to Appendix A Figure A1.

*1.4. Problem Statement*

The main concern is a DC attack used in IoT devices by intruders to discover the cryptographic keys of an AES algorithm. An AES can suffer from a DC attack [1,9]. For instance, a DC attack has been applied experimentally on a d Mini-AES algorithm [14]. The experiment revealed more than 50 percent of the secret key. An AES has been attacked using an algebraic DC attack to crack the secret key [15]. The fundamental principle that the DC attack adventured was the high probability of particular events of plaintext pair differences and ciphertext pair differences generated in the last round, which has been conducted in the study done by [7]. Lacko-Bartosova [16] presented a DC attack of a two-round AES with a complexity approximation of a three-round AES attack. Lacko-Bartosova [16] has also indicated that a DC attack depends on recommendation particular bitwise text differences. Grassi [5] has attacked five-round AES using a DC attack and "multiple-of-8" rule. Tunstall [1] says the first attack is a four-round AES DC attack subjected to a differential that supplemented a significant probability. The second attack is a five-round AES Square attack that needs a time complexity of 237.5 during the encryption process and 28 pairs of ciphertexts to break an AES secret key [1].

An AES is being used on IoT devices even though it is attackable. For instance, IoT devices use an AES algorithm to encrypt and transfer the encrypted data to the next layer of security known as Message Queuing Telemetry Transport protocol [4]. Message Queuing Telemetry Transport protocol is an ISO standard (ISO/IEC PRF 20922) to transmit encrypted data. On the recipient side, the encrypted data was being decrypted using an AES algorithm [4]. The VMware SD-WAN Edge comprises VMware SD-WAN Dynamic Multipath Optimization (DMPO) and an Extensive Application Recognition as IoT devices aggregated on reoccurring links used to direct traffic across optimal links [17]. Additionally, traffic is being led to other VMware SD-WAN Edges of different departments, private data centers, universities, and offices, using an AES for secure communication [17]. Sophia et al. [18] have indicated that the health department is the swelling concern of the patients worldwide. An e-healthcare Remote Clinical Sensor Network is supported in collecting the vital body information of individual terminals using sensors as IoT devices. The suggested technique is on the principles of implementing a secured key and being encoded by an AES [18].

With all this information, the concern of this study is to recure an AES from the DC attacks and secure all IoT devices and data using an AES algorithm. A DC attack can ruin the whole security of IoT devices and consumers if it is not duly analyzed. Little has been

done to improve the number of output bits on the S-Boxes to resist a DC attack [1,9]. This study focuses on resolving a DC attack on an AES.

The newly generated 32-output bits S-Boxes are utilized to block DC attacks of an AES detected on IoT devices. A KDM function makes a new 32-Bit S-Box suitable for the new Modified AES Algorithm and confuses the attacker since it comprises many mathematical modulo operators. Additionally, most mathematical modulo operators are irreversible. Additionally, the novel approach of employing a KDM function and converting the $8 \times 8$ S-Boxes to be the $8 \times 32$ S-Boxes successfully prevents DC attacks on an AES algorithm. A KDM function is a new mathematically generated, named, and designed for this study. A KDM function was never developed, defined, or utilized before by any researcher except for in this study.

```c
int KDM_Function(state32hold)
{
 int i,j;
 uint64_t TempState =0, Muthavhine=4294967296, T=0,V=0, X=0;
 uint64_t Khumbelo = 041760427607,
 Difference= 040035532523;
 uint64_t Mod[6]= {256604724, 40037230360,
  7779667,4294968531,0273,4};
 for(i=0,j=0;i<4&&j<4;i++,j++)
 {
 T = state32hold[j][i] * (bool)(state32hold[j][i]/ Muthavhine);
 V = Muthavhine * (bool)(Muthavhine / state32hold[j][i]);
 state32hold[j][i] = T+V;
 }
 for(i=0,j=0;i<4&&j<4;i++,j++)
 {
 if (state32hold[j][i] > Muthavhine)
 { Khumbelo =    (Mod[0] ^ Khumbelo);
   Difference=  (Mod[2] ^ Muthavhine) % Khumbelo;
   Muthavhine =     (Mod[2] ^ Difference) % Mod[3];
 }
  else
    {
    Muthavhine = (state32hold[j][i]<<Mod[4]) % Khumbelo;
    Khumbelo = (state32hold[j][i] <<Mod[5]) % Difference;
    Difference = ((state32hold[j][i]% Khumbelo) <<Mod[4]);
    Khumbelo = (Muthavhine ^ Difference)% Mod[2];
    Difference = ((Muthavhine % Khumbelo)^ Mod[0]);
    Muthavhine = (Khumbelo ^ Difference)% Muthavhine ;
    }//Modular operators (%) make variable to be unknown,
  } // invisible and irreversible to an intruder.
  for(i=0,j=0;i<4&&j<4;i++,j++)
  { while (Khumbelo)
   {
    TempState = (~state32hold[j][i]) & Khumbelo;
    state32hold[j][i]=_abs64(state32hold[j][i] ^ Khumbelo);
statehold[j][i]=(state32hold[j][i]/(Mod[2]^Muthavhine))^Mod[4];
    Khumbelo = (TempState << 1);
   }
    TempState = Khumbelo ^ TempState;
    Khumbelo = Khumbelo % Muthavhine;
   }
}
```

**Figure 6.** A KDM function to make a new 32-S-Box suitable for the modified AES algorithm.

### 1.5. Theoretical Confirmation of DC Attack on AES

The DC attack has initially been presented on the AES-128 decreased to five rounds by Biham and Keller [23,24]. That was later developed by Cheon et al. [25] to discover six rounds utilizing 291.5 preferred plaintext pairs and time complexity of 2122. For AES-192 and AES-256, Raphael and Phan [26] achieved to attack both AES-192 and AES-256 reduced to seven rounds [24]. The DC attack needed 292 (AES-192) and 292.5 (AES-256) chosen-plaintext pairs with time complexities of 2186 (AES-192) and 2250.5 (AES256) respectively [24]. Currently, the best DC attack filters AES-128 up to six rounds [24]. For both AES192 and AES-256, the best DC attack so far succeeds in breaking through seven rounds [24].

Lacko-Bartosova [16] used the DC attack on two rounds of AES with the calculation of complexity for a three-round AES attack. Given the DC attack, which was based on discovering apparent bitwise differences of the secret key. The data complexity of the defined DC attack was 227, where 8 bits of the private key were recovered [16].

Jakimoski and Desmedt [27] used a related-key DC attack to the 192-bit secret key modification of AES. Jakimoski and Desmedt [27] also indicated that although any 4-round DC attack had at least 25 active bytes of the secret key. The intruder could invent a 5-round related-key DC attack that isolated and cracked 15 active bytes of the private key and revealed a 6-round key with 2106 plaintext/ciphertext pairs and complexity 2112 [27]. Jakimoski and Desmedt [27] indicated that the attack could be enhanced using a truncated DC attack. In that case, the required number of plaintext/ciphertext pairs could be 281, which was about 286 of computational complexity. Utilizing impossible related-key DC attack, Jakimoski and Desmedt [27] claimed to break 7-rounds with a computational complexity of 2116 and 2111 plaintext/ciphertext pairs. The attack on 8-rounds required a complexity of about 2183 encryptions and 288 plaintext/ciphertext pairs [27].

Hu and He [28] utilized a new property of MixColumns Transformation and constructed a new 4-round impossible DC attack path. Hu and He [28] added 1-round and 3-round possible DC attack paths before and behind the path, respectively. Additionally, Hu and He [28] constructed a new 7-round impossible DC attack path. Hu and He [28] utilized the path to analyze 64-bit initial keys of 7-round AES-192, and that analysis method required 271 pairs of selected plaintexts, about 272 memory cells, and about 2135 encryption and decryption computation. Finally, they recovered the secret keys [28].

Rouquette and Solnon [9] indicated that based on the complete distribution ratio and complexity that occurred, Mini-AES algorithms were vulnerable to a DC attack [9]. The best DC attack characteristic is the DC attack characteristic utilizing a single active S-Box with the distribution ratio of 8/16 [9]. Rouquette and Solnon [9] used the distribution ratio of 8/16 as the probability of guessing the secret key.

This study has found no denial from the above information that AES is being attacked using the DC attack on different rounds. Other information is detailed in Section 2 (the literature review section) of this study. Additionally, more experimental data are explained in Section 4 of this paper for experimental confirmation of the DC attack on AES done in this study.

### 1.6. The Objective of the Study

An AES can suffer from a DC attack [1,9]. This study aims to solve the problem of a DC attack used in IoT devices by intruders to discover the cryptographic keys of an AES algorithm. Additionally, the study aims at solving the problem of using a KDM function and the newly generated the 32 output bits S-Boxes to generate a new Modified AES Algorithm that confuses and blocks the attacker from applying the DC attack.

## 2. Literature Review

Tunstall [1] presented an experimental intricacy of an AES DC attack. The results showed that most attacks used the same approach and application but used incompatible models. Tunstall [1] drew the improved attacks suggested in other literature reviews using

different models on differential fault and DC attacks. The attack was a four-round AES DC attack subjected to a differential that supplemented a significant probability. Javed et al. [3] indicated that cell phones and wireless networks as IoT devices are found using AES for confidentiality, integrity, and availability of data. Additionally, the packet filtering and patches method of cellular phones and wireless networks was found using an AES for security [3].

Heys [7] conducted an experiment driven by the basic Substitution-Permutation Network algorithm of an AES. The presentation gave a comprehensive understanding of the DC attack as applied to the algorithm. It was helpful since an Advanced Encryption Standard (AES) had been based on the basic Substitution-Permutation Network structure [1,7]. Furthermore, experimental results from the DC attacks were conferred as evidence of accepting the idea as outlined. Even though the first plan of DC attack was on DES [1,16], however, the extensive applicability of DC attacks to several other cryptographic algorithms thickened the superiority of DC attack techniques in the security inspection of all cryptographic algorithms [1,7,9,22,29]. Cryptologists developed technology based on techniques explicitly targeted at DC circumvention [6,7,9]. That was evident, for instance, in the Rijndael cipher, the cryptographic cipher nominated to be the prospective standard [9,14,26]. Rokan et al. [11] indicated the use of sensors as IoT devices that connect embedded-subsystem using networks. An AES was found helping IoT sensors' communication security, which was being established in various IoT devices such as intelligent energy-grids, Machine to Machine (M2M) communications, buildings, and data computing devices [11].

Z'aba and Maarof [10] applied a differential cryptanalytic attack on a reduced number of rounds, and the complexity was compared to that of an exhaustive research attack. An exhaustive research is an attack that probes every key possibility value of a cryptographic algorithm [5,6,10,26]. Consequently, an exhaustive research attack was treated as the upper bound attack in cryptography [6,10]. Z'aba and Maarof [10] reviewed other existing cryptanalytic attacks on an AES. However, the focus was on DC attacks. Z'aba and Maarof [10] indicated that the superiorities of attacks were grounded in the principle of a DC attack. For instance, the impossible differential attack was utilized on the MixColumns transformation of an AES [10,29]. If a pair of plaintext varied only in one byte, then the reduced four rounds ciphertext of an AES would never be the same in the ciphertext byte positions: $(0,0), (1,3), (2,2), (3,1), (0,1), (1,0), (2,3),(3,2), (0,2), (1,1), (2,0), (3,3)$ nor $(0,3), (1,2), (2,1), (3,0)$ [10]. Wrong key bytes were removed if the impossible event exists [10]. The impossible differential attack was initially presented on an AES-128 after being reduced to five rounds by Biham and Keller [5,6,10,10]. Z'aba and Maarof [10] indicated that a cryptologist called Cheon later improved the impossible differential attack up to six rounds utilizing a time complexity of 2122 and 291.5 chosen plaintext. For the AES-256 and AES-192. Z'aba and Maarof [10] indicated that a cryptologist called Phan achieved attacking the seven rounds reduce AES. The attack needed 292.5 (AES-256) and 292 (AES-192) chosen plaintexts with time complexities of 2250.5 (AES-256) and 2186 (AES-192), respectively. The impossible differential attack worked better on AES-128 up until six rounds [10]. Applied to AES-256 and AES-192, the impossible differential attack was hitherto accomplished to discover the key up to seven rounds [10]. The impossible related-key differential is an attack that uses the key scheduling of a cryptographic algorithm [10,22,29]. The impossible related-key differential inspected the deportment of an AES by applying a variant but related keys. The impossible related-key differential attack was unrelated to the inner structure, and the number of rounds [10]. The combination of the impossible related-key attack and the impossible DC attack gave good results [10]. When an impossible related-key differential attack was applied on an AES-192, then the variety of two attacks was capable of breaking up to seven rounds of an AES utilizing 2111 plaintext/ciphertext pairs and time complexity of 2116 [10]. An AES was found being used on IoT devices even after it was broken or attacked. Munoz et al. [12] indicated that IoT boards and IoT edge as examples of IoT devices were found using an AES algorithm for data security.

Grassi [5] indicated that at Eurocrypt 2017, an initial secrete key differentiator for five-round AES depended on the "multiple-of-8" rule had been conferred. Despite the fact that a secrete key differentiator permits to differentiate a random AES permutation, it is evidently rather hard to apply a key-recovery attack different than an exhaustive research, using such a differentiator [5,8,14]. An AES was found being used on IoT devices even after it was broken or attacked. Alshammari et al. [30] indicated sensor nodes recognized with their IoT limited abilities, and implementing software based on the truly security protocols caused the subject to be cumbersome. Assuring security in sensor nodes as IoT devices, communications were found being encrypted using an AES algorithm [30].

Lacko-Bartosova [16] presented DC of two-round AES with a complexity approximation of a three-round AES attack. Lacko-Bartosova [16] also indicated that DC attacks depend on recommendation, particular bitwise text differences. Complexity data described the differential attack of 227, where a subkey byte was retrieved. Lacko-Bartosova [16] described a DC attack that was initially introduced at the crypto conference in 1990 by E. Biham and A. Shamir as a cryptanalysis attack applied on DES [16].

Heys [7] defined the fundamental principle that DC adventured a high probability of particular events of plaintext pair differences and ciphertext pair differences generated in the last round. It was a chosen-plaintext cryptanalysis attack, which means the modus operandi was to select plaintext, and ciphertext was consequently calculated to recover the secret key. An AES was found being used on IoT devices even after it was broken or attacked. Saraiva et al. [13] indicated that the PRISEC module of the UbiPri middleware was one of the IoT devices that were found using an AES to secure data privacy and protection. Simmons [15] stipulated that a Simplified AES was developed to educate students about the fundamental understanding of an AES. An AES was designed in such a way that the DC was not valueless on simplified AES [15]. An algebraic DC attack is an approach that exploits modern mathematical equation solvers to attack ciphers such as an AES [8,9,15,26]. Simmons [15] indicated that there had been a few allegations that an AES and a DES were vulnerable to algebraic DC attacks. Simmons [15] utilized an algebraic DC attack to crack a simplified AES. Algebraic DC attack was a imaginably convincing attack on symmetric-key block algorithms [10,15,22]. An algebraic DC attack started by creating a quite substantial non-linear structure of polynomial equations in terms of input plaintext bits, input key bits, and output ciphertext bits and then attempted to crack that structure by using an imaginably convincing equation-solving application [5,15,22]. The variable number, the polynomial numbers, and the polynomial degrees, the power of the mathematical equations, and memory and the speed of the computer being utilized resolved whether the infrastructure could be able to reveal the secret key bits [15]. Despite the fact, an AES was still cracked by an algebraic DC attack and a simplified AES was quickly broken by the DC attack, even though an AES was still openly used [10,15,22]. Rekha and P. Saravanan [31] indicated that edges such as IoT devices were accessible through the internet and were found using an AES to secure the accumulated data collected from the sensors located in the field.

Gemellia [14] presented the experimental results of a DC attack applied on a Mini-AES algorithm. To give the experimental results, Gemellia [14] implemented the key eradication for differential characteristics which yielded the lowest and highest characteristics and the probability as a correlation. Depending on the propagation ratio amount and complexity obtained by Gemellia [14], Mini-AES algorithms were defenseless to DC attack. The first-rate differential characteristic was by utilizing a single active S-Box of Mini-AES algorithm that yielded the propagation ratio of $8/16 = 0.5$ [14]. The LoRaWAN protocol was being used for low energy consumption [32]. The particular LoRaWAN protocol comprised large networks with many IoT devices to secure bi-directional communication for machine-to-machine (M2M), smart city, and industrial applications using an AES algorithm for secure data communication [32].

Ankele et al. [6] showed that the Substitution Permutation Networks were one of the essential functions used to design cryptographic algorithms such as AES and DES.

Ankele et al. [6] applied a DC attack on a three-round Substitution Permutation Network. Ankele et al. [6] had utilized a 16-bit plaintext, a 16-bit ciphertext, and selected the first row of a third DES' S-Box of DES for the importance of an S-Box and ShiftRows transformation to permute bytes in an AES for Substitution Permutation Networks. Consequently, Ankele et al. [6] had revealed a 12-bit key of a 16-bit key from the final round of an AES, DES, and Skinny algorithm using the DC attack method. Farooq et al. [33] indicated that a tremendous amount of data contains information stored in health monitoring systems, intelligent cars, industrial plants, and intelligent buildings, as IoT devices were being encrypted using an AES.

Khurana and Kumar [8] presented a multiset of state vectors with an integral 'n' representing the number of bytes in the ciphertext and plaintext. The steps Khurana and Kumar [8] demonstrated to finding the variants, distinguishing, and revealing the key using a DC attack would considerably help the attacks of cryptographic algorithms such as DES and AES. Nandan et al. [34] indicated that the Xilinx nexys 4 Artix 7 –FPGA board and Xilinx ISE hardware suite using telosB sensor mote as an IoT device to sense room temperature. Data collected by telosB sensor mote were found to be encrypted using an AES algorithm [34]. Amrita et al. [21] indicated that the DC utilized similarities that exist between differences in the input and output of a building block of an algorithm such as Mixcolumn in an AES. In the response of a cryptographic algorithm such as an AES, plaintext pairs with established differences were scrutinized [6]. Amrita et al. [21] used a DC attack to exploit plaintext pairs and expose the probabilities to various subkeys bits. Results indicated that an AES was then vulnerable to various attacks such as DC [21,22,26]. Amrita et al. [21] indicated that applicable improvements were accessible which, when accurately implemented, could resolve these vulnerabilities at a high level. Other methods such as hybrid attacks, man-in-the-middle attacks, and Denial of Services attacks were making slow progress, but no successful attacks had been recorded [10,16,21]. Amrita et al. [21] indicated that evolutions showed that an AES would not survive the expectancy of the conventional algorithm suite recognized for confidential applications. Additionally, Amrita et al. [21] indicated that evolutions could cause an AES an irrelevant preference for confidential and extensive applications. Nonetheless, modernized secure strategical communications tools such as IoT devices use programmable cryptographic algorithms such as an AES [8,15,21].

Muthavhine and Sumbwanyambe [35] indicated that an AES was found being used on IoT devices to secure sensors and encrypt contactless intelligent cards.

Rijmen [22] showed that cryptologists did not have enough time to develop a robust 128-bit cryptographic algorithm such as an AES. After intensive research, Rijmen [22] found that the theoretical security level of algorithms like the AES candidates would be $2^{100}$ or less if approximately 5 to 10 years would be spent in the effort of severe DC attacks. Clarity as a design principle was challenged on the risk of failure and resistance against DC attacks [15,21,22]. Additionally, there were curiosities about the anxiety of mitigating analysis and mitigating DC attacks. Rijmen [22] indicated that mitigations relied on the opinion, and many programming languages were not yet supporting the Finite field as a building block of the AES candidates. The S-Boxes were still a challenge to program in hardware platforms. Alimi et al. [36] indicated that the DASH7 Alliance protocol provided various layers of security in protocols Low Power Wide Area Networks (LPWAN) and was being used as an activator network protocol embedded with a wireless sensor network. Securing communication being established by the DASH7 Alliance protocol depended on an AES-128 encryption scheme [36]. An AES-128 encryption algorithm was found driving security in LoRa as an IoT device [36].

Sophia et al. [18] indicated that the security of the health department is a growing concern of patients worldwide. The e-healthcare Remote Clinical Sensor Network supported collecting the vital body information of individual terminalsusing sensors as IoT devices.

Jithendra and Shahana [29] indicated that the security of an algorithm was typically evaluated through the operation of different models of cryptanalysis methods. A

cryptanalysis method employing impossible differentials for a cryptanalysis attack was observed to be a feasible method for retrieving the secret keys of an algorithm such as an AES [8,15,22,29]. Related keys were applied to increase rounds to apply unacceptable conditions to minimize impossible cryptanalysis attack complexity [29]. Jithendra and Shahana [29] introduced a new related-key and reduced round attack to measure an AES-192 strength. Most of the attacks stopped at the seven-round attack presented earlier as the better method [14,21,26,29]. Jithendra and Shahana [29] created an eight-round attack utilizing a new relative key, which exposed the secrete keys with the lowest time complexity. The VMware SD-WAN Edge was composed of VMware SD-WAN Dynamic Multipath Optimization (DMPO) and an Extensive Application Recognition as IoT devices aggregated on reoccurring links used to direct traffic across optimal links [17]. Additionally, traffic was being led to other VMware SD-WAN Edges of different departments, private data centers, universities, and offices, using an AES for secure communication [17].

Rouquette and Solnon [9] proposed Constraint Programming models to solve DC attack problems on a cryptographic algorithm such as an AES. The models were more effective than devoted approaches even though the program was cumbersome compared to simple models and showed no scalability, and it was essential to introduce improved constraints contemplated from cryptographic properties [6,22,26,29]. Rouquette and Solnon [9] introduced a global constraint that refined the mathematical modeling steps in an understandable way and reformed the efficiency to improve implementation. Rouquette and Solnon [9] also studied an AES complexity, introduced propagators, and practically analyzed them on single-key and related-key cryptanalysis attack problems for Midori and an AES algorithm [9]. The results showed that the global constraint permitted the calculation of Maximum Differential Characteristics (MDCs) at a higher speed than advanced models (which were cumbersome to design a program) for single-key and related-key cryptanalysis attacks on Midori, and additionally, for single-key cryptanalysis attacks on an AES. Although, a related-key cryptanalysis attack on an AES failed to solve the two biggest instances of an AES-192 within an acceptable quantity of time [9]. Ahamed et al. [4] indicated that Secure Hashing Algorithm-256 and AES-256 were proposed to fulfill the security of IoT devices. The data collected from IoT devices were found to be initially encrypted using an AES-256 with an SHA-256 symmetric key, and finally, encrypted data was being produced [4]. IoT devices transferred that encrypted data to the next layer of security known as Message Queuing Telemetry Transport protocol, an ISO standard (ISO/IEC PRF 20922) being used to transmit encrypted data. On the recipient side, the encrypted data was being decrypted [4].

## 3. Research Methodology

The primary objective research of this study is to defend an AES algorithm discovered on IoT devices against a DC attack. This study replaced an original 8-Bit-output S-Box and the inverse Box of an AES algorithm with the newly generated 32-Bit-output S-Boxes. A unique mathematical function called KDM is developed for the suitability of the newly generated 32-Bit-output S-Boxes. The newly generated 32-Bit-output S-Boxes are inserted on an AES algorithm to get more a desirable encryption and decryption process with the protection against a DC attack. A KDM function is used to make the new 32-Bit S-Box suitable for the new Modified AES Algorithm and confuse the attacker since it comprises many mathematical modulo operators. Additionally, most mathematical modulo operators are irreversible. A new modified AES algorithm is developed after embedding the newly generated 32-Bit-output S-Boxes and a KDM function in an AES's infrastructure. In this study, the newly modified AES algorithm, with the newly generated 32-Bit-output S-Boxes and a KDM function, is coined M_AES. The mode of operation of M_AES is very distinctive and is related to an original AES algorithm since the strength, the encryption process, and the resistance of the DC attacks is more substantial than an original AES algorithm found on IoT devices. The research is conducted as follows:

1.  An original AES algorithm is found in IoT devices (such as cellphones, smart cards, Machine to Machine (M2M), and sensors);
2.  The accuracy of an original AES algorithm is validated and examined using test vectors given in the literature review;
3.  All the executed procedures on an original AES algorithm during DC attacks are experimented and verified using C++;
4.  Both an original 8-Bit-output S-Box and the inverse S-Box of an AES are converted to the newly created 32-Bit-output S-Boxes;
5.  The new KDM function is embedded in an original AES algorithm infrastructure using C++. Refer to Figure 6;
6.  All other functions using an S-Box and the inverse of 8-Bit-output from an original AES algorithm are changed to use a KDM function with the newly 32-Bit-output S-Boxes as an input of a KDM function. For instance, if

$$Output \; = \; C \; = \; S_i(P). \tag{1}$$

Note: $S_i(P)$ Equation (1) uses an 8-Bit-output S-Box. Equation (1) is substituted using Equation (2).

$$KDM_function(S_i(P), Khumbelo), \tag{2}$$

$S_i(P)$ Equation (2) uses a new 32-Bit-output S-Box because an AES S-Box and its inverse are converted to give the new 32-Bit-output S-Boxes;

7.  The possibility for the DC attacks is reconstructed on the M_AES algorithm. If the DC attacks are still successful after a newly 32-Bit-output S-Box and a KDM function has been embedded, and if it is furthermore achievable, steps three and four are re-conducted;
8.  If DC attacks are prevented in steps three, four, and five, then a new M_AES algorithm embedded with a newly 32-Bit-output S-Box and a KDM function is accepted as a M_AES algorithm.

The research methodology performed a Difference-Distribution Table more obstreperous to block the attackers from discovering AES's keys after DC attacks are applied. The security of the M_AES algorithm depends on the size of the S-Boxes output bits and a KDM function. The originals output bits of an AES's S-Box and its inverse are low (8-Bit). It is simple for intruders to attack such a kind of algorithm. A newly generated 32-Bit-output S-Box and its inverse are employed to substitute all the 8-Bit-output S-Boxes and improve the size of output bits from 8 to 32-bits for the M_AES algorithm that is used to improve the output bits robust against DC attacks. Experiments showed that a new 32-Bit-output S-Box and its inverse worked successfully to block DC attacks. At the same time, a KDM function is used to make a new 32-Bit S-Box suitable for the new Modified AES Algorithm and confuse the attacker since it comprises many mathematical modulo operators. Additionally, most mathematical modulo operators are irreversible. The research methodology is outlined, utilizing the schematic diagram in Figure 7. The results successfully prevented the construction of the Difference-Distribution Table and produced a complex process to conduct DC attacks on the M_AES algorithm (refer to Figure 8). Comparing Figures 1 and 8, the difference is a new 32-Bit-output S-Box, the inverse 32-Bit output S-Box, and a KDM function. Consequently, the M_AES algorithm is found to be repellent to the DC attacks. Refer to Figure 8.

An AES's S-Box and its inverse were discovered to be 8x8, indicating that they have 8-Bit-inputs and 8-Bit-outputs, respectively. The research found that it is simple to construct a Difference-Distribution Table utilizing these descriptions of the S-Boxes. For instance, back to our example, a $4 \times 4$ S-Box illustrated in Table 1 yielded a Difference-Distribution Table of $2^4 \times 2^4$ illustrated in Table 3 with high-probability components of detecting secret key bits. Commonly, if an S-Box has X-Bit of inputs and Y-Bit of output, then its Difference-Distribution Table, when created, will be a $2^X \times 2^Y$ matrix. Hence, the Difference-Distribution Table illustrated in Table 3, is shown to be $2^4 \times 2^4$. In this study, the

C++ code is written to create a Difference-Distribution Table of $2^4 \times 2^4$ illustrated in Table 3 using Equation (2). The code proved to be simple for attacking any algorithm using a $4 \times 4$ S-Box illustrated in Table 1. Additionally, the code indicated that it is used to construct the Difference-Distribution Table of $2^8 \times 2^8$, using $8 \times 8$ AES Box and its inverse is defined in Figure 2.

To prevent the DC attacks, a new 32-Bit-output S-Box and its inverse are generated to replace the $8 \times 8$ AES Box, and its inverse is defined in Figure 2.

For instance, an AES S-Box in Figure 2 is replaced with a new 32-Bit output of an AES S-Box. An AES inverse S-Box in Figure 2 is replaced with the new 32-Bit output of an AES inverse S-Box. A KDM function is constructed for the suitability of a new 32-Bit-output S-Box and its inverse in a new M_AES algorithm. A KDM function makes a new 32-Bit S-Box suitable for the new Modified AES Algorithm and confuses the attacker since it comprises many mathematical modulo operators. Additionally, most mathematical modulo operators are irreversible. A new 32-Bit S-Box is resistant to the DC attacks (refer to Figure 8). Comparing Figures 1 and 8, the M_AES algorithm shown in Figure 8 is resistant to DC attacks compared with the AES algorithm shown in Figure 1.

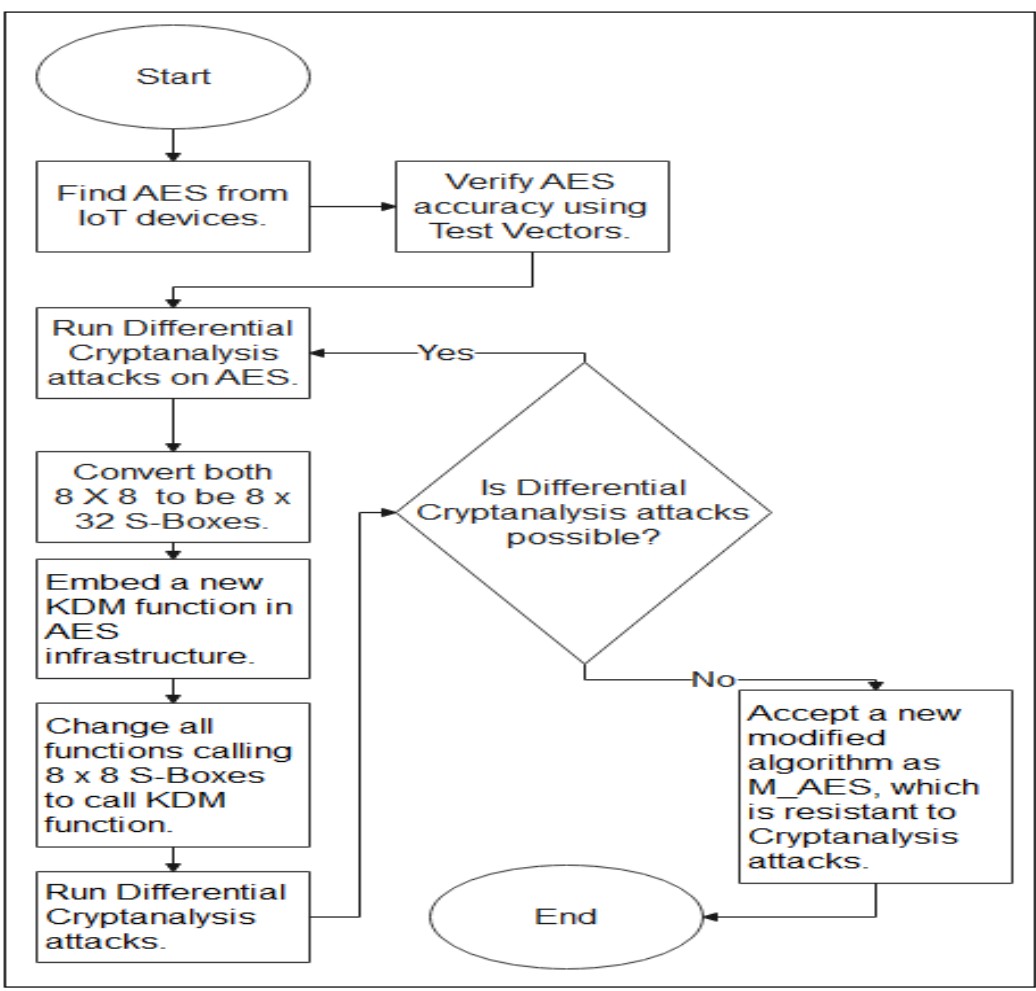

**Figure 7.** Flowchart or schematic diagram of the research methodology.

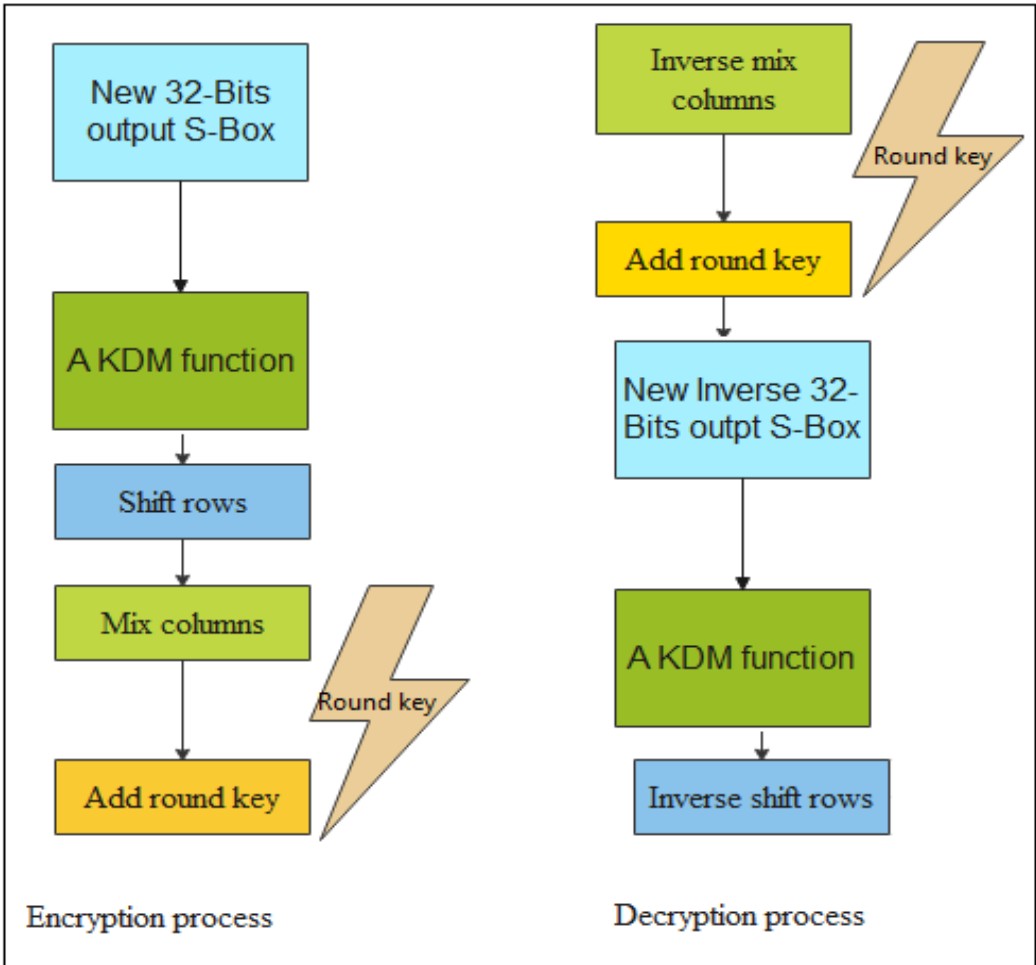

**Figure 8.** New modified AES (M_AES) algorithm with the encryption and decryption process.

## 4. Results and Analysis

On an AES, results showed that a DC attack was possible. The main components that made all the possibilities of a DC attack were the size of the S-Boxes. An S-Box of an AES was $8 \times 8$, indicating a 8-Bit input and a 8-Bit output. The Difference-Distribution Table discovered that it was straightforward to create the Difference-Distribution Table utilizing an $8 \times 8$ AES S-Box.

The study wrote a C++ program to create the Difference-Distribution Table of a $4 \times 4$, a $6 \times 4$, a $8 \times 8$, and a $8 \times 32$ S-Box. The validation of code was tested using a simplified DES's S-Box of $4 \times 4$ given in Table 1, a $6 \times 4$ DES S-Box given in [37] page 12 and 13, an $8 \times 8$ AES S-Box given in Figure 2 and a new generated $8 \times 32$ S-Box of M_AES algorithm. The aim of validating the code was to verify the correctness of the written C++ experimental output Difference-Distribution Table compared to the theoretical outputs. Figure 2 shows that no DDT was feasible to be constructed due to the high percentage of memory needed to build DDT. No DDT and no DC attack occurred according to the definition and the procedure of a DC attack.

The C++ Difference-Distribution Table of a $4 \times 4$ S-Box. The entities were the same as in Table 3. Therefore, the C++ Difference-Distribution Table of $4 \times 4$ was executing the correct results. The time taken to execute the C++ Difference-Distribution Table of a $4 \times 4$ S-Box was 0.2815 s. Note that the Difference-Distribution Table of a $4 \times 4$ S-Box is a matrix of $2^4 \times 2^4 = 16 \times 16$ matrix with 256 entities. For the C++ Difference-Distribution Table of $6 \times 4$, and the entities were the same as in the theoretical Difference-Distribution Table given in [37] pages 12 and 13. Therefore the C++ Difference-Distribution Table of $6 \times 4$ was executing the correct results. The time taken to execute the C++ Difference-Distribution

Table of a 6 × 4 S-Box was 1.2100 s. Note that the Difference-Distribution Table of a 6 × 4 S-Box is a matrix of $2^6 \times 2^4 = 64 \times 16$ matrix with 1024 entities.

The experiment continues on an 8 × 8 AES S-Box. Note that the Difference-Distribution Table of an 8 × 8 AES S-Box is a matrix of $2^8 \times 2^8 = 256 \times 256$ matrix with 65,536 entities. To display an entire visible 256 × 256 matrix, 5 pages are needed.

The experiment continues on a newly generated 8 × 32 S-Box of M_AES algorithm. The program crashed after 3 h before the Difference-Distribution Table was executed. No machine or computer could compute the Difference-Distribution Table of $2^8 \times 2^{32} = 256 \times 4{,}294{,}967{,}296$ matrix, expected to contain 1,099,511,627,776 entities. Without a Difference-Distribution Table, it was impossible to conduct a DC attack on a newly generated 8 × 32 S-Box of M_AES algorithm.

A Difference-Distribution Table of a 4 × 4 S-Box had the first entity of integer 16, which is $(2^4)$ since an S-Box needed four bits as the highest parameter. A number 16 is a byte donated as 00010000 in binary. If each entity of a 4 × 4 S-Box Difference-Distribution Table is treated as a byte, then memory needed to construct a 4 × 4 S-Box Difference-Distribution Table was 8 bits × 256 = 256 bytes. Note that 256 is the number of entities displayed on a 4 × 4 S-Box Difference-Distribution Table. A machine or computer can easily handle 4096 bytes.

Difference-Distribution Table of a 6 × 4 S-Box had the first entity of integer 64, which is $(2^6)$ since a S-Box needed six bits as the highest parameter. Number 64 is a byte donated as 001000000 in binary. If each 6 × 4 S-Box Difference-Distribution Table entity is treated as a byte, then the memory needed to construct a 6 × 4 S-Box Difference-Distribution Table was 8 bits × 1024 = 1024 bytes. Note that 1024 is the number of entities displayed on a 6 × 4 S-Box Difference-Distribution Table. A machine or computer can easily handle 1024 bytes.

A Difference-Distribution Table of an 8 × 8 S-Box had the first entity of integer 256, which is $(2^8)$ since an S-Box needed 8 bits as the highest parameter. A number 256 is a word composed of 2 bytes donated as 0000000100000000 in binary. If each 8 × 8 S-Box Difference-Distribution Table entity is treated as a word, then the memory needed to construct an 8 × 8 S-Box Difference-Distribution Table is 16 bits × 65,536 = 131,072 bytes. Note that 65,536 is the number of entities displayed on an 8 × 8 S-Box Difference-Distribution Table. A machine or computer can handle 131,072 bytes.

From the above calculations, the study expected that the Difference-Distribution Table of an 8 × 32 S-Box would have the first integer entity as 4,294,967,296, which is $(2^{32})$ since an S-Box needed 32 bits as the highest parameter. 4,294,967,296 is a triple-word composed of 5 bytes donated as 0000000010000000000000000000000000000000 in binary. If each 8 × 32 S-Box Difference-Distribution Table entity were treated as a triple-word, then the memory needed to construct an 8 × 32 S-Box Difference-Distribution Table would be 40 bits × 1,099,511,627,776 = 5,497,558,138,880 bytes. Note that 1,099,511,627,776 was an expected number of entities displayed on an 8 × 32 S-Box Difference-Distribution Table. A machine or computer could not easily handle a computation memory of 5,497,558,138,880 bytes of each entity. Hence the C++ Difference-Distribution Table of an 8 × 32 S-Box program crashed before execution. All the findings were given in Tables 4–6. Comparison of the findings were explained graphically using Figures 9–11.

Difference-Distribution Table of an AES S-Box was a table $2^8$ rows × $2^8$ columns with great probabilities of figuring a key. The C++ program was written to create the Difference-Distribution Table of an 8 × 8 AES S-Box. After investigating the procedure, the results verified that it was feasible to attack an AES algorithm utilizing the Difference-Distribution Table. The newly generated 32 output bits S-Boxes were used on an AES found on a IoT devices to prevent a DC attack. Additionally, the novel approach of changing the 8 × 8 S-Boxes to be the 8 × 32 S-Boxes successfully blocks the DC attacks on an AES algorithm. A KDM function is a new mathematically developed function, coined, and used purposely in this study. A KDM function was never produced, defined, or utilized before by any researcher except in this study. A KDM function was used to make the new 32-Bit S-Boxes

suitable for the new Modified AES Algorithm and confuse the attacker since it comprised many mathematical modulo operators. Additionally, most mathematical modulo operators were irreversible.

A C++ program was written to create the Difference-Distribution Table of an $8 \times 32$ AES S-Box. The code crashed before constructing a Difference-Distribution Table of a new S-Box, which was assumed to be a $2^8 \times 2^{32}$ matrix. The results showed that it was infeasible to create a Difference-Distribution Table of a new $8 \times 32$ AES S-Box with an output of 32-Bit because a computer has limited memory compared to the required memory to construct a Difference-Distribution Table of a new S-Box. The first trial was to apply an array of a $2^{32} = 4,294,967,296$ size; the results showed that input $2^8 = 256$ also had to be added. This prevented creating a Difference-Distribution Table of an $8 \times 32$ AES S-Box due to memory constraints required by the computer.

The program of creating a Difference-Distribution Table of a new S-Box failed before the construction of a Difference-Distribution Table due to memory needed to run, display and execute a $256 \times 4,294,967,296$ matrix by a computer. Calculation of $2^{32} \times 256$ required more than $2^{64}$ memory allocation, which is impracticable when using a computer. The research also validated that it was impractical to create a table or any matrix of $256 \times 4,294,967,296$ due to memory constraints allocated in a computer. The boundaries of memory were $2^{64}$ in Microsoft (Hp) and Macintosh (Apple) computers, which caused a Difference-Distribution Table difficulty for the DC attack. To get the probabilities of calculating a key of 32-Bit output S-Box was impractical. Therefore, the results prevented Difference-Distribution Table construction using the newly 32-Bit output S-Boxes and a KDM function that was a new mathematically developed function, coined, and used purposely in this study. A KDM function was never produced, defined, or utilized before by any researcher except in this study. A KDM was generated for the suitability of newly developed the 32-Bit-output S-Boxes in a freshly modified AES algorithm. The study used a KDM function to make the new 32-Bit S-Box suitable for the new Modified AES Algorithm and confuse the attacker since it comprises many mathematical modulo operators. Additionally, most mathematical modulo operators are irreversible. For further information about a KDM function, refer to Figure 6.

It was found that no Difference-Distribution Table resulted in a DC attack. Consequently, in this study, the results intensified the protection of an AES against a DC attack.

To confirm that all methods of the DC attack using a Difference-Distribution Table were conducted, the C++ executable file of a Difference-Distribution Table defined in Table 3. The study conducted the experimental DC attack on Simplified-DES and AES. In this paper, the study explains only portions of a round of the practical DC attack. The rest is the repetition of the same process on each round to complete an entire attack.

### 4.1. Experimental Confirmation of the DC Attack on Simplified-DES

The study first conducted the DC attack on a Simplified-DES to verify the attack before attacking AES. Consider the following mathematical functions: $ciphertext_1 = plaintext_1 \oplus key$ By using the difference of a ciphertext pair of ciphertext, the calculation would have dropped out the *key* required, giving us no knowledge about the *key*: $ciphertext_1 \oplus ciphertext_2 = plaintext_1 \oplus key \oplus plaintext_2 \oplus key \ ciphertext_1 \oplus ciphertext_2 = plaintext_1 \oplus plaintext_2$.

The above function shows that the difference between the plaintext is equivalent to the difference between the ciphertext.

Note that Simplified-DES is not a linear algorithm function. Therefore, the difference between plaintext is not the same as the difference between ciphertext. Considering Simplified-DES, the difference in a plaintext pair for a specific difference of a ciphertext pair is determined by the *key* value. From the Difference-Distribution Table given in Table 3, $plaintext_1 \oplus plaintext_2 = \Delta P \ ciphertext_1 \oplus ciphertext_2 = \Delta C$ With the guidance of the Difference-Distribution Table the study got the output and input values from Table 3. For instance, when $\Delta P = 12$ and $\Delta C = 3$, the possible of *key* occurrence is two. That

is $\Delta P = 6 \oplus 10$ or $\Delta P = 10 \oplus 6$. Therefore possible two input pairs are $(6, 10)$ and $(10, 6)$. Consider input pair $(6, 10)$, then $plaintext_1 = 6$, $plaintext_2 = 10$ and assume then $ciphertext\_1 = 3$ and $ciphertext_2 = 0$ therefore $\Delta C = 3$. If the input difference of a $4 \times 4$ S-Box is denoted by $H = H_1 \oplus H_2$, let us assume that $H_1 = plaintext_1 \oplus key$ and $H_2 = plaintext_2 \oplus key$. From the above analysis, the *key* has no influence on the input difference value because is the same constant value, therefore: $\Delta P = H = 6 \oplus 10 = 12$ meaning $H = 12 = 4 \oplus 8$ if $\Delta C$ is assumed to be equal to 0 using the Difference-Distribution table. H is a pair of $(H_1, H_2) = (4, 8)$. that is $H = 12 = 4 \oplus 8$ $key = H \oplus \Delta P$ therefore $key = H_1 \oplus plaintext_1$ and $key = H_1 \oplus plaintext_2$. Substituting the values $key = H_1 \oplus plaintext_1 = 4 \oplus 6 = 2$ and $key = H_1 \oplus plaintext_2 = 4 \oplus 10 = 14$. Alternatively $key = H \oplus \Delta P$ therefore $key = H_2 \oplus plaintext_1$ and $key = H_2 \oplus plaintext_2$. Substituting the values $key = H_2 \oplus plaintext_1 = 8 \oplus 6 = 14$ and $key = H_1 \oplus plaintext_2 = 8 \oplus 10 = 2$. Therefore two possible *key* values are found, that is, 2 and 4. Each *key* is tested to give the value of $\Delta C$, the one that gives the same value of a pair is the right *key*. In this case, two is the right tested *key*. Therefore $key = 2$. With this information, the study confirmed that the Simplified-DES is crackable using the DC attack. The DC attack managed to crack both two rounds of a Simplified DES using a ciphertext pair of $2^{10}$ with a time complexity of $2^{16}$. Then, the same procedure was used on DES.

Table 7 shows that no DDT was feasible to be constructed due to the high percentage of memory needed to build DDT. No DDT and no DC attack occurred according to the definition and the procedure of a DC attack. Refer to Table 7 and Figure 12.

### 4.2. Experimental Confirmation of the DC Attack on DES

The study used an input pair $\Delta P$ to a DES S-Box as $(1, 35)$ where $\Delta P = Plaintext_1 \oplus Plaintext_2 = 1 \oplus 35$, therefore $\Delta P = 34$. Suppose, $\Delta C = D$. $\Delta P = 34$, regardless of the *key* value because $H_1 = Plaintext_1 \oplus key$ and $H_2 = Plaintext_2 \oplus key$, therefore $H = H_1 \oplus H_2$ $H = (Plaintext_1 \oplus key) \oplus (Plaintext_2 \oplus key)$ $H = Plaintext_1 \oplus Plaintext_2$ $H = \Delta P$. Also $H_1 = \Delta P \oplus key$ and $key = H \oplus DeltaP$. Using the Difference-Distribution Table, the possible *key* occurrence is 8, which is $\{07, 11, 17, 1D, 23, 25, 29, 33\}$.

If the same procedure was repeated when input pair $\Delta P$ to a DES S-Box as $(21, 15)$, but still keeping $\Delta P = 34$ since $21 \oplus 15 = 34$, and change $\Delta C = 3$ instead of using $\Delta C = D$. Using the Difference-Distribution Table, the possible *key* occurrence is 6, which is $\{00, 14, 17, 20, 23, 34\}$. The accurate *key* value should visible in both of these groups: $\{07, 11, 17, 1D, 23, 25, 29, 33\}$ and $\{00, 14, 17, 20, 23, 34\}$ which $\{17, 23\}$ either 17 or 23 is the right *key* value. Each *key* is tested to give the value of $\Delta C$, the one that gives the same value of pair is the right *key*. In this case, 17 is the right tested *key*. The DC attack managed to crack all 16 rounds of DES using a ciphertext pair of $2^{14}$ with a time complexity of $2^{58}$. Then, the same procedure was used on AES. Refer to Table 7 and Figure 12.

### 4.3. Experimental Confirmation of the DC Attack on AES

The study determines a difference in a byte. A byte has 8 bits, then $2^8 = 256$ possible ciphertexts have to be generated. Once all 256 possible ciphertexts have been developed, the last subkey can be verified using the Difference-Distribution Table hypotheses. Hypotheses testing is accomplished by examining conditions on the final subkey byte-by-byte. The study analyses that the input pair to the final round is equal to zero. The calculation returns the accurate subkey. The study also expects one extra wrong hypothesis byte-by-byte, given that a random distribution has an input pair equal to zero with a probability of $1/256$ from the Difference-Distribution Table hypotheses. The analysis resulted in an anticipated total number of key assumptions for the final subkey of $2^{16}$. The DC attack managed to crack 7 rounds out 10 using a ciphertext pair of $2^{92}$ with a time complexity of $2^{186}$. Refer to Table 7 and Figure 12.

### 4.4. Experimental Confirmation of the DC Attack on M_AES

M_AES utilized a new 32-bit S-box which failed to execute the C++ Difference-Distribution Table from different machines and computers due to the memory constraints of different machines and computers. No machine or computer could compute the Difference-Distribution Table of $2^8 \times 2^{32} = 256 \times 4{,}294{,}967{,}296$ matrix, expected to contain 1,099,511,627,776 entities. Without a Difference-Distribution Table, it was impossible to conduct the DC attack on a newly generated 8 × 32 S-Box of M_AES algorithm. No round out of 16 was cracked using the DC attack due to a new 32-bit output S-Box, which blocked the construction of the Difference-Distribution Table due to machine memory constraints. Refer to Table 7 and Figure 12.

Analysis of how Table 3 was theoretically created was investigated and written in practical C++ code for validation, testing, and verification (refer to Table 3) Table 3 had the same probability entities. Table 3 was a theoretical Difference-Distribution Table, which was used to verify and confirm that the investigation of creating a Difference-Distribution Table was conducted with all the methods of the DC attack on an AES.

The code was also applied to both AES and M_AES to test whether the DC attack was possible. All the findings were given in Tables 4 and 5. Table 4 showed all the construction of the Difference-Distribution Table before and after a Novel Approach of using the 32-Bit S-Boxes were applied. The study used a KDM function to make a new 32-Bit S-Box suitable for the new Modified AES Algorithm and confuse the attacker since it comprises many mathematical modulo operators. Additionally, most mathematical modulo operators are irreversible. Table 5 showed the results of key bits discovery before and after a novel approach of using a KDM function and the 32-Bit S-Boxes was applied.

In this study, M_AES was resistant to the DC attack and was constructed using the new 8 × 32 S-Boxes. The study used a KDM function to make a new 32-Bit S-Box suitable for the new Modified AES Algorithm and confuse the attacker since it comprises many mathematical modulo operators. Additionally, most mathematical modulo operators are irreversible. New M_AES was adequate to decrypt and encrypt successfully after using a KDM function and the new 8 × 32 S-Boxes. The code of newly M_AES is available on request. The C++ code showed that the DC attack was possible to a standard AES on several rounds before using a KDM function and the new 8 × 32 S-Boxes. However, after using a KDM function and the new 8 × 32 S-Boxes as a novelty, the C++ code showed that the DC attack was prevented successfully on M_AES. Additionally, it was difficult to construct a Difference-Distribution Table of $2^{32}$ rows and columns matrix due to the memory limitation of a computer. All the findings were given in Tables 4–6. Comparison of the findings were explained graphically using Figures 9–11.

**Table 4.** Results of feasibility of creating Difference-Distribution Table before and after a novel approach of using a KDM function and the 32-Bit S-Boxes were applied.

| Name of Algorithms | Before a Novel Approach of Using a KDM Function and the 32-Bit S-Boxes Were Applied | After a Novel Approach of Using a KDM Function and the 32-Bit S-Boxes Were Applied |
|---|---|---|
| AES | The construction of a Difference-Distribution Table was feasible. | The construction of a Difference-Distribution Table was infeasible due to the memory limitation of the computer. |

**Table 5.** Results of key bits discovery before and a novel approach of using a KDM function and the 32-Bit S-Boxes were applied.

| Name of Algorithms | Before a Novel Approach of Using a KDM Function and the 32-Bit S-Boxes Were Applied | After a Novel Approach of Using a KDM Function and the 32-Bit S-Boxes Were Applied |
|---|---|---|
| AES | The key was discovered in many rounds. | No key bits were discovered or detected in all rounds of an AES. |

**Table 6.** Results of creating a Difference-Distribution Table (DDT).

| The Size of the S-Box | Time Taken (in Seconds) to a Create Difference-Distribution Table (DDT) | Number of Entities Required | Memory (in Bytes) Needed |
|---|---|---|---|
| 4 × 4 | 0.2815 | 256 | 256 |
| 6 × 4 | 1.2100 | 1024 | 1024 |
| 8 × 8 | 23.6800 | 65,536 | 131,073 |
| 8 × 32 | ∞ | 1,099,511,627,776 | 5,497,558,138,880 |

**Table 7.** Results of a differential cryptanalysis attack.

| Name of Algorithm | Number of Rounds Attacked during a DC Attack Process in % |
|---|---|
| Simplified DES (S-DES) | 2 out 2 or 100% |
| DES | 16 out 16 or 100% |
| AES | 7 out 10 or 70% |
| M_AES | 0 out 10 or 0% |

In cryptography, the Avalanche Effect is the acceptable property of algorithms [38]. If one input bit is changed (flipped), the output bits have to change significantly. Such a slight modification in either the plaintext or the key should create an extreme difference in the ciphertext in robust algorithms [38]. The Avalanche Effect is advanced to get a procedure called the Strict Avalanche Criterion (SAC) to test the encryption strength of the algorithm [39]. The SAC is fulfilled if a single input bit, either the plaintext or the key, yields the change of ciphertext output bits of 50% probability [39]. This study conducted the Avalanche Effect on S-DES, DES, AES, and M_AES to get SAC. The results showed that the AES and a newly generated M_AES algorithm had a better SAC property than S-DES and DES since the Avalanche Effect of M_AES on both key and plaintext were approximately a 50% probability compared to S-DES and DES (refer to Table 8)

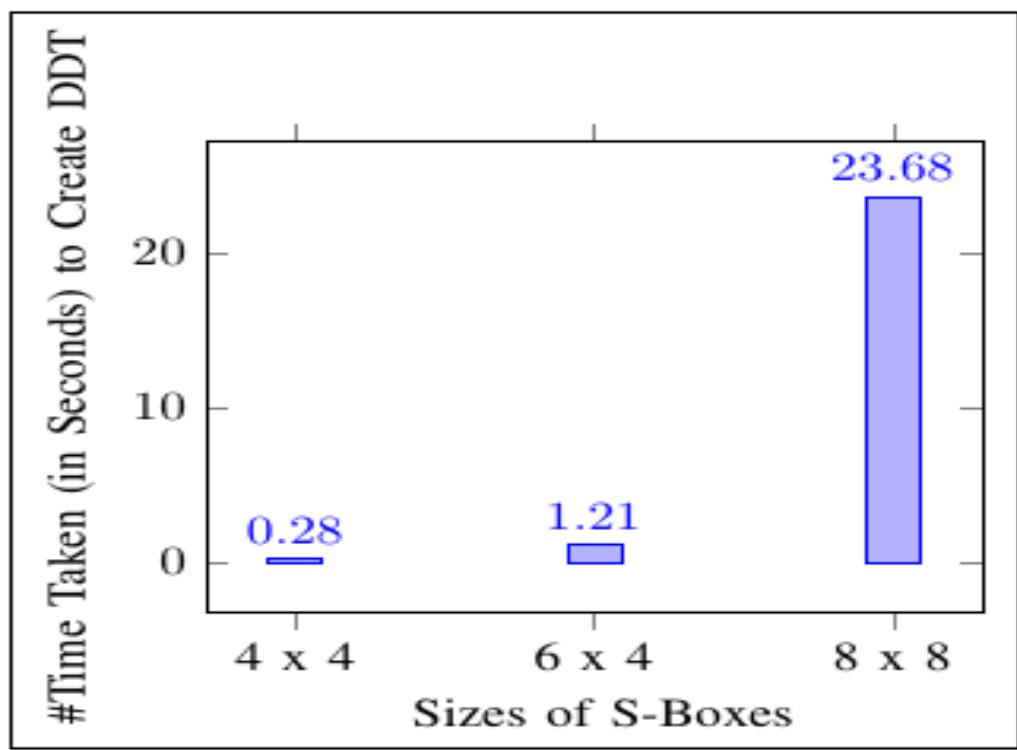

**Figure 9.** Experimental time taken to create a DDT.

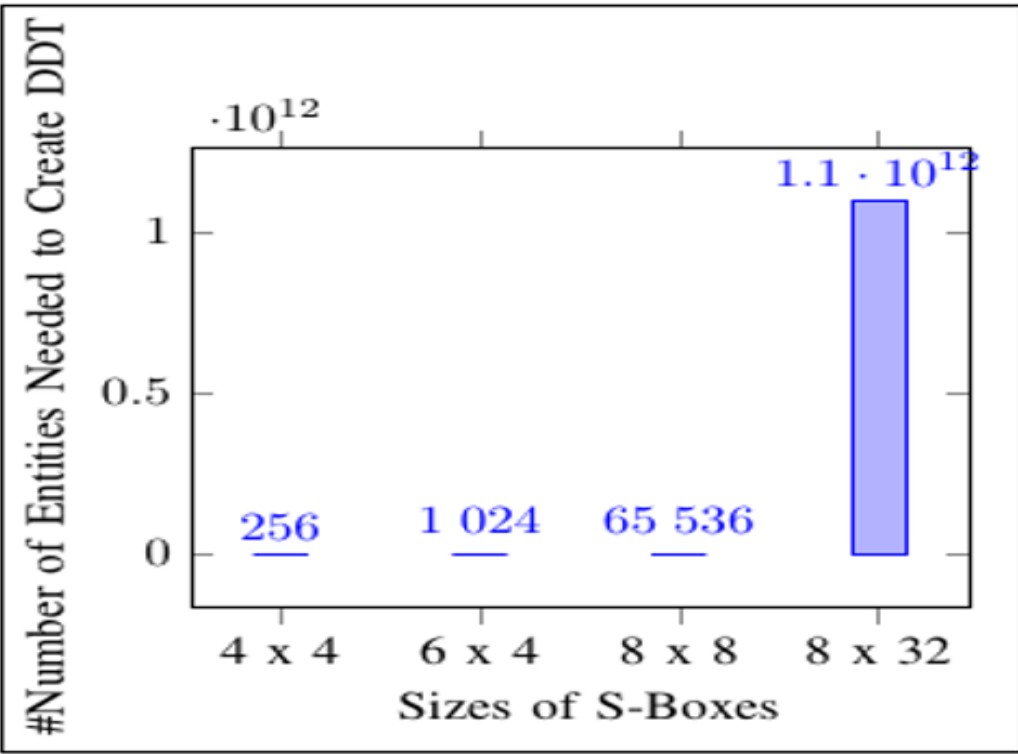

**Figure 10.** Experimental number of entities to create a DDT.

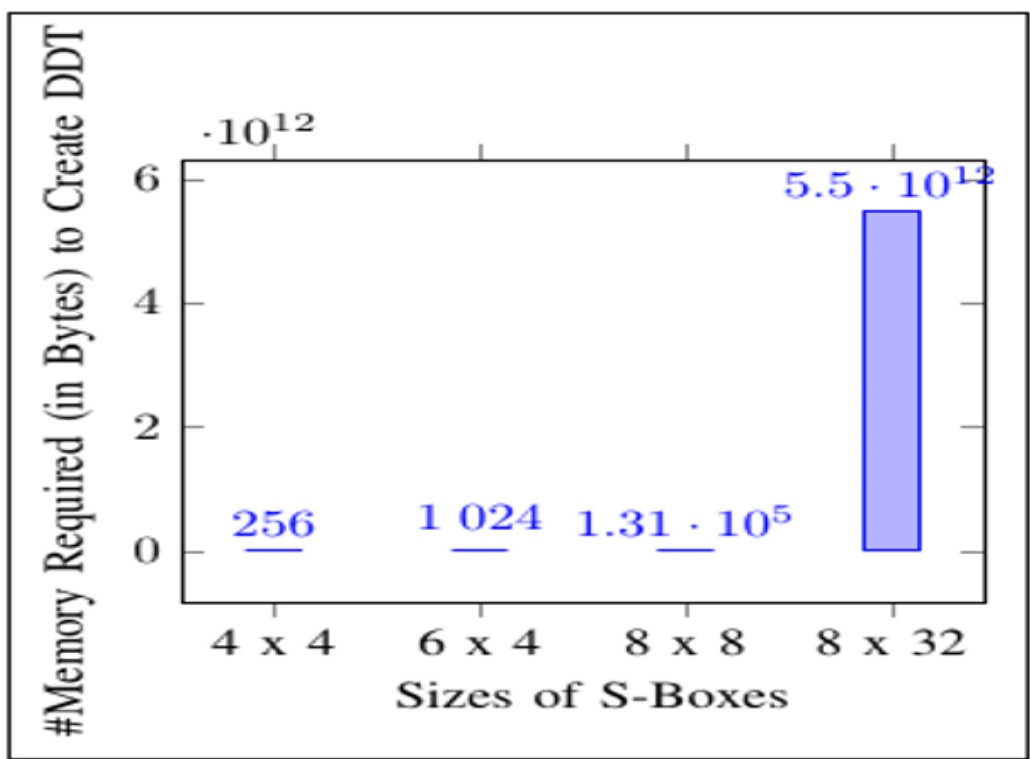

**Figure 11.** Experimental memory required to create DDT.

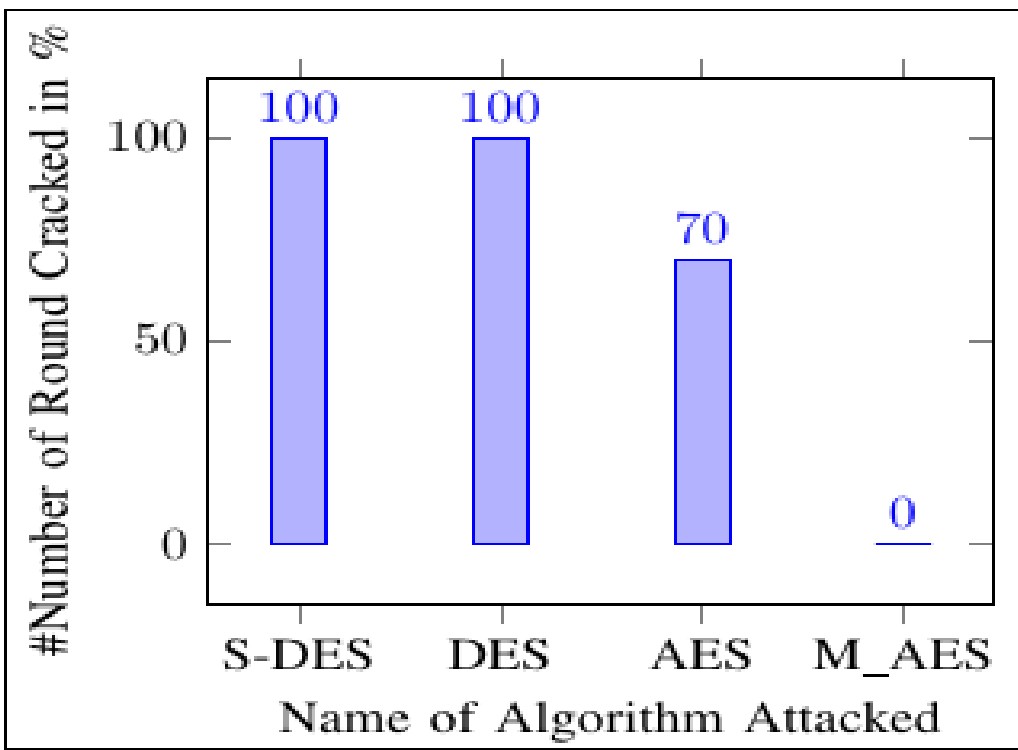

**Figure 12.** Experimental number of rounds cracked during a differential cryptanalysis attack.

In cryptography, a tool called the Strict Avalanche Criterion (SAC) is applied to decide if an algorithm is adequate to be powerful or not using the interpretation of the output of the Avalanche Effect. The SAC is satisfied if, whenever a 1-Bit input is flipped, the output bits should vary with approximately 50% (the range between 45% and 55%) of the Avalanche Effect probability. For example, take a sample of algorithms, say A and B,

which have 15% and 25% of the Avalanche Effect probability, respectively. Both A and B fail the criterion since their Avalanche Effect is considerably less than approximately 50%. Additionally, if an algorithm has 85%, then according to the SAC definition, that distinct algorithm fails the criterion since 85% is considerably greater than approximately 50% (the range between 45% and 55%) of the Avalanche Effect probability. Therefore, an algorithm with roughly 50% has a higher encryption strength than other algorithms with considerably less and more significance than approximately 50%. The results showed that the AES and a newly generated M_AES algorithm had a better SAC property than S-DES and DES since the Avalanche Effect of M_AES on both key and plaintext were approximately a 50% probability compared to S-DES and DES (Refer to Table 8)

All algorithms (S-DES, DES, AES, and M_AES) managed to encrypt and decrypt the same image, but encrypted images were not the same (refer to Figure 13).

**Table 8.** The avalanche effect of the key and plaintext bit that were flipped.

| Name of Algorithm | Plaintext Avalanche Effect in Percentage | Key Avalanche Effect in Percentage |
|---|---|---|
| Simplified DES (S-DES) | 25 | 25 |
| DES | 60.4003 | 44.2138 |
| AES | 50.0488 | 50.2807 |
| M_AES | 49.9023 | 50.2807 |

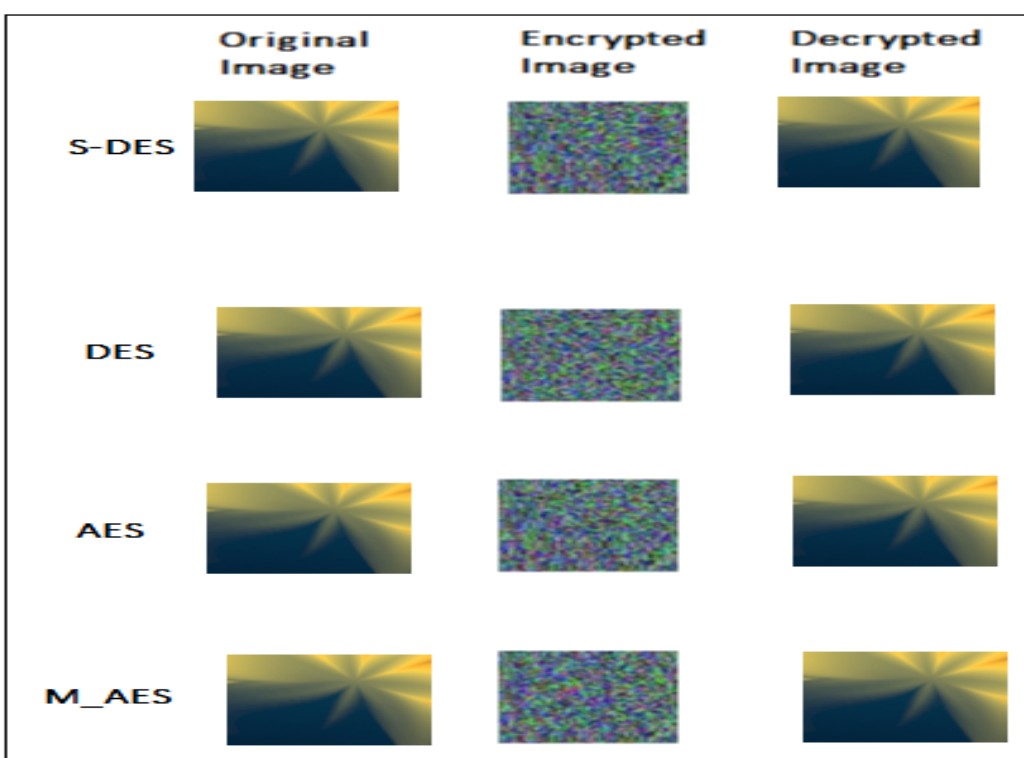

**Figure 13.** Image Encryption of All Algorithm.

## 5. Conclusions and Future Work

The study confirms that an AES used on IoT devices is vulnerable to DC attacks. The sizes of a AES S-Box and the inverse are 8 × 8. The study reveals that these S-Boxes are the first building blocks applied during a DC attack because the output size is smaller than 32-Bit. An S-Box of 8 × 8 gives a Difference-Distribution Table of $2^8$ rows × $2^8$ columns, which is a sound probability table for an attacker to conduct the DC attack.

This study confirms that it is convincing to prevent all schemes from administering the DC attack on an AES algorithm commonly utilized on IoT devices by employing a novel approach by using the newly generated 32-Bit S-Boxes AES. The study used a KDM function to make a new 32-Bit S-Box suitable for the new Modified AES Algorithm and confuse the attacker since it comprises many mathematical modulo operators. Additionally, most mathematical modulo operators are irreversible.

The study examines how to make a cryptanalysis attack more challenging to create and to make it more difficult for the intruder to calculate the keys of an AES after using a KDM function and the newly generated 32-Bit S-Boxes. Results confirm that the security of any algorithm such as an AES relies on the size of the output of the S-Boxes. If the size of the output bits of an S-Box is small, it is easy to reveal the secret key of that particular algorithm. The study confirms that a novel approach using a KDM function and the newly generated 32-Bit S-Boxes successfully confuses and prevents the DC attacks, respectively. The results managed to stop the Difference-Distribution Table construction successfully. Additionally, the results were cumbersome while administering the DC attack.

In the future, the succeeding research will be on how to block IoT devices against other kinds of attacks such Boomerang attacks using a KDM function and the 32-bit output S-Boxes.

**Author Contributions:** Supervision, M.S.; writing—original draft, K.D.M. All authors have read and agreed to the published version of the manuscript.

**Funding:** This research received no external funding.

**Institutional Review Board Statement:** Not applicable.

**Informed Consent Statement:** Not applicable.

**Data Availability Statement:** The data presented in this study are available in article.

**Conflicts of Interest:** The authors declare no conflict of interest.

**Appendix A**

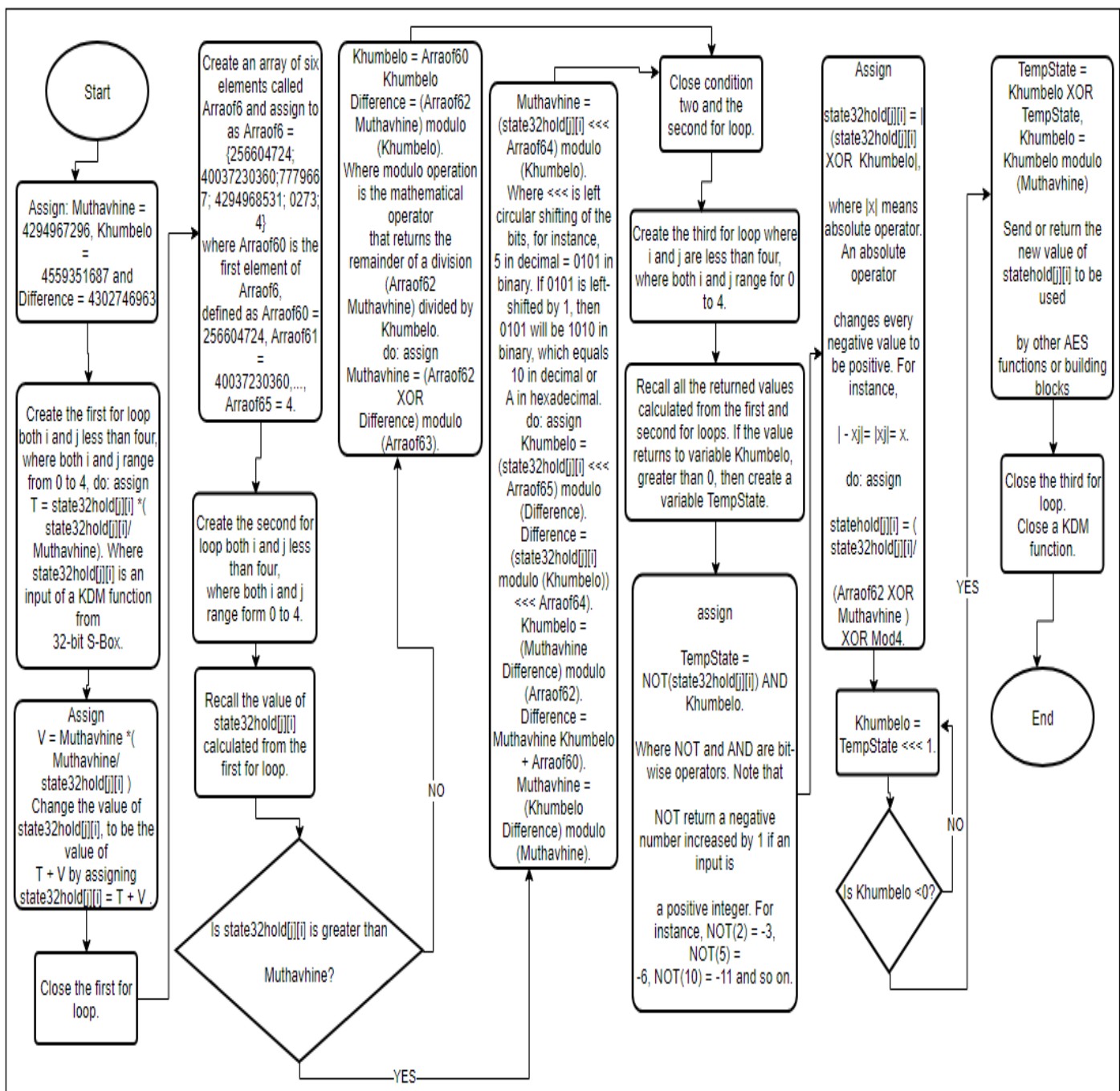

**Figure A1.** Flowchart of a KDM function.

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
