# Peer review of "Preventing Differential Cryptanalysis Attacks Using a KDM Function and the 32-Bit Output S-Boxes on AES Algorithm Found on the Internet of Things Devices"

_cryptography, doi:10.3390/cryptography6010011_

Round 1

Reviewer 1 Report

This paper focus on how to preventing differential cryptanalysis attacks on the AES Algorithmon AES, and 
a novel KDM Function and the 32-Bit  output S-Boxes are proposed. It has academic reference significance for traditional cryptography researchers. There are only some minor comments on the manuscript. 
1. The English grammar of every sentence and the logic flow problem should be re-check. 
2. Many singular nouns lack the article a in front of them, such as:  "S-Box is ...."->"An S-Box is ....". 
3. "changes a string of plaintext (input) into 4X4 matrix"-->"changes a string of plaintext (input) into a 4×4 matrix".
4. "c0 = x, which is a row, and 00 = y, which is a column"-->"c0 = x, which is a row number, and 00 = y, which is a column number".
5."8-bits" should be "8-bit" or "8 bits".  "anew" -->"a new" in line 1581.
6."inverse an AES S-Box"-->"an inverse AES S-Box".
7. In line 582: "Phan [?] achieved", [?]. 
8. Many "X" should be changed to  "×", such as 4 X 4, 8X8, 256 X 256, ...
9. Since the KDM function comprises many mathematical modulo operators. Additionally, most mathematical modulo operators are irreversible. I have a question is how to ensure that the ciphertext can be decrypted accurately in the Modified AES Algorithm? 

Author Response

COMMENTS FROM REVIEWER ONE

  1. The English grammar of every sentence and the logic flow problem should be re-check.  Two authors and Grammarly software check grammar
  1. Many singular nouns lack the article a in front of them, such as:  "S-Box is ...."->"An S-Box is ....". I confirm that I have rectified the article
  2. "changes a string of plaintext (input) into 4X4 matrix"-->"changes a string of plaintext (input) into a 4×4 matrix". I confirm that I have rectified the X to be x
  3. "c0 = x, which is a row, and 00 = y, which is a column"-->"c0 = x, which is a row number, and 00 = y, which is a column number". I confirm that I have rectified the column to column number.
  4. ."8-bits" should be "8-bit" or "8 bits".  "anew" -->"a new" in line 1581. I confirm that I have rectified this.
  5. "inverse an AES S-Box"-->"an inverse AES S-Box". I confirm that I have rectified this.
  1.  In line 582: "Phan [?] achieved", [?]. I confirm that I have rectified this.
  2. Many "X" should be changed to  "×", such as 4 X 4, 8X8, 256 X 256, ... I confirm that I have rectified this.
  3.  Since the KDM function comprises many mathematical modulo operators. Additionally, most mathematical modulo operators are irreversible. I have a question is how to ensure that the ciphertext can be decrypted accurately in the Modified AES Algorithm? See the video attached

COMMENTS FROM REVIEWER TWO

1) The objective of this paper is not clear. Please, write a paragraph (at the end of "Introduction" to describe the objective. I confirm that I have rectified this.

2) The literature review is too long. It is a paper, NOT a thesis. Try to summarize it. I confirm that I have rectified that I have reduced the literature review.

3) The words (Differential Cryptanalysis) are repeated many many times. Use abbreviation for easier presentation. I confirm that I have rectified this.

4) In Sec. III, "Research methodology", the steps are clear. But, use present verbs NOT past! It is your work. Ex: The accuracy of an original AES algorithm was validated and examined using................ use (is) instead of (was).... allover the procedure. I confirm that I have rectified this.

5) I prefer to delete Figs. 7 and 8. Instead, write a summary of both figures (if important). The same remark for Figs. 11, 12, 13, (18 to 25). I confirm that I have rectified this.

6) I miss the discussion of Figs. 26, 2, 28... what is the benefit or advantage (or disadvantage) of the results represented in the 3 figures. I confirm that I have rectified this.

7) Same remark (like 6) for Tables VI, VII, VIII. I confirm that I have rectified this.

Figure 2 shows an 8x8 AES Box and it inverse. Both are converted to be the 8x32 to prevent the construction of DDT. Note that, No DDT no DC attack according to the definition AND the procedure of DC attack.

Table V1 Shows that no DDT was feasible to be constructed due to high percentage of memory needed to construct DDT. No DDT no DC attack according to the definition AND the procedure of DC attack.

Table VII shows that no DC attack succeeded on our newly generated M\_AES algorithm. On M\_AES algorithm no round is attacked compared to other algorithms in the table. Infinity,  referred to M\_AES algorithm in the Table, means many plaintext and ciphertext were used but failed to crack the key bits

Table VIII. In cryptography, a tool called Strict Avalanche Criterion (SAC) is applied to decide if an algorithm is adequate to be powerful or not using the interpretation of the output of Avalanche Effect. SAC is satisfied if, whenever a 1-Bit input is flipped, the output bits should vary with approximately $50\%$ (the range between $45\%$ to $55\%$) of Avalanche Effect probability \cite{bn27}. For example, take a sample of algorithms, say A and B, which have $15\%$ and $25\%$ of Avalanche Effect probability, respectively. Both A and B fail the criterion since their Avalanche Effect is considerably less than approximately $50\%$. Additionally, if an algorithm has  $85\%$, then according to SAC definition, that distinct algorithm fails the criterion since $85\%$ is considerably greater than approximately $50\%$ (the range between $45\%$ to $55\%$). Avalanche Effect probability. Therefore, an algorithm with roughly $50\%$ has high encryption strength than other algorithms with considerably less and more significance than approximately $50\%$. The results showed that the AES and a newly generated M\_AES algorithm had a better SAC property than S-DES and DES since the Avalanche Effect of  M\_AES on both key and plaintext were approximately $50\%$ probability compared to S-DES and DES. Refer to Table \ref{table:139}.

8) The multiplication symbol is (x..... insert,  symbol) NOT the letter X)... correct both text and figures axis. I confirm that I have rectified this.

9) The "Abstract" misses at least one (main) numerical value. I confirm that I have rectified this.

10) The "Conclusion" is too long. When writing, do not repeat sentences, and correct "has confirmed" to "confirms".  I confirm that I have rectified I have reduced conclusion.

11) Typos are to be revised. Two authors and Grammarly software check grammar. 

12) When the reference is a conference (Ex: 7, 17, ......), write the place where the conference was held (city + country) and the exact date (as possible) day/month/year. I confirm that I have rectified this.

Reviewer 2 Report

Thank you for your efforts.

I have some comments that must be considered in the modified manuscript. 

-----------------------------------------------------------------

1) The objective of this paper is not clear. Please, write a paragraph (at the end of "Introduction" to describe the objective.

2) The literature review is too long. It is a paper, NOT a thesis. Try to summarize it.

3) The words (Differential Cryptanalysis) are repeated many many times. Use abbreviation for easier presentation.

4) In Sec. III, "Research methodology", the steps are clear. But, use present verbs NOT past! It is your work. Ex: The accuracy of an original AES algorithm was validated and examined using................ use (is) instead of (was).... allover the procedure.

5) I prefer to delete Figs. 7 and 8. Instead, write a summary of both figures (if important). The same remark for Figs. 11, 12, 13, (18 to 25). 

6) I miss the discussion of Figs. 26, 2, 28... what is the benefit or advantage (or disadvantage) of the results represented in the 3 figures.

7) Same remark (like 6) for Tables VI, VII, VIII. 

8) The multiplication symbol is (x..... insert,  symbol) NOT the letter X)... correct both text and figures axis.

9) The "Abstract" misses at least one (main) numerical value.

10) The "Conclusion" is too long. When writing, do not repeat sentences, and correct "has confirmed" to "confirms".

11) Typos are to be revised.

12) When the reference is a conference (Ex: 7, 17, ......), write the place where the conference was held (city + country) and the exact date (as possible) day/month/year.

Author Response

(The authors gave the same response as above.)
